# Spatiotemporal recruitment of RhoGTPase protein GRAF inhibits actomyosin ring constriction in *Drosophila* cellularization

**Swati Sharma, Richa Rikhy***

Biology, Indian Institute of Science Education and Research, Pune, India

**Abstract** Actomyosin contractility is regulated by Rho-GTP in cell migration, cytokinesis and morphogenesis in embryo development. Whereas Rho activation by Rho-GTP exchange factor (GEF), RhoGEF2, is well known in actomyosin contractility during cytokinesis at the base of invaginating membranes in *Drosophila* cellularization, Rho inhibition by RhoGTPase-activating proteins (GAPs) remains to be studied. We have found that the RhoGAP, GRAF, inhibits actomyosin contractility during cellularization. GRAF is enriched at the cleavage furrow tip during actomyosin assembly and initiation of ring constriction. *Graf* depletion shows increased Rho-GTP, increased Myosin II and ring hyper constriction dependent upon the loss of the RhoGTPase domain. GRAF and RhoGEF2 are present in a balance for appropriate activation of actomyosin ring constriction. RhoGEF2 depletion and abrogation of Myosin II activation in Rho kinase mutants suppress the *Graf* hyper constriction defect. Therefore, GRAF recruitment restricts Rho-GTP levels in a spatiotemporal manner for inhibiting actomyosin contractility during cellularization.

***For correspondence:**
richa@iiserpune.ac.in

**Competing interests:** The authors declare that no competing interests exist.

## Introduction

Metazoan embryogenesis involves a variety of cell shape changes during cytokinesis, cell migration and tissue morphogenesis (*Agarwal and Zaidel-Bar, 2019*; *Heer and Martin, 2017*; *Kumar et al., 2015*; *Lecuit and Lenne, 2007*; *Levayer and Lecuit, 2012*; *St Johnston and Ahringer, 2010*; *Yam et al., 2007*). The cortical actomyosin activity is orchestrated with plasma membrane shape remodeling to generate localized forces to drive cell shape dynamics (*Heer and Martin, 2017*; *Jodoin et al., 2015*; *Martin et al., 2009*; *Mason et al., 2013*; *Munjal et al., 2015*; *Murrell et al., 2015*). The intensity and directionality of force generated are dependent on spatiotemporal regulation of Myosin II activity (*Heisenberg and Bellaïche, 2013*; *Rauzi et al., 2010*). Myosin II activation by phosphorylation is achieved by Rho kinase (*Amano et al., 2010*; *Kasza et al., 2014*; *Levayer and Lecuit, 2012*; *Vasquez et al., 2014*). Rho-GTP, a member of the small GTPases belonging to the Ras superfamily, activates Rho kinase (*Prudnikova et al., 2015*). These molecular switches undergo cycling between GDP-bound-inactive and GTP-bound-active states (*Agarwal and Zaidel-Bar, 2019*). This Rho-GTP and Rho-GDP cycling is catalyzed by guanine nucleotide exchange factors (GEFs) and GTPases-activating proteins (GAPs) (*Bos et al., 2007*; *Van Aelst and D'Souza-Schorey, 1997*). Rho-GEFs lead to the production of Rho-GTP, whereas RhoGAPs promote hydrolysis of the GTP to generate Rho-GDP (*Jacobs and Hall, 2005*). The relative expression and spatiotemporal distribution of RhoGEFs and RhoGAPs are critical in fine-tuning the levels of Rho-GTP and extent of Myosin II activation in different cellular processes (*Agarwal and Zaidel-Bar, 2019*; *Chircop, 2014*; *Mulinari and Häcker, 2010*; *Wu and Priya, 2019*).

*Drosophila* embryogenesis is an excellent model system to study the molecular mechanisms regulating actomyosin-driven contraction in a spatiotemporal manner. The *Drosophila* embryo begins

development as a syncytium (*Foe et al., 2000*; *Foe and Alberts, 1983*; *Schejter and Wieschaus, 1993a*). Transition from the syncytial to cellular blastoderm occurs at the embryo cortex in the interphase of cycle 14 by cleavage to separate each nucleus by plasma membrane boundaries in a process called cellularization (*Mazumdar and Mazumdar, 2002*; *Schejter and Wieschaus, 1993a*; *Sullivan and Theurkauf, 1995*). The leading edge of the invaginating plasma membrane between adjacent nuclei occurs as an infolding termed as furrow canal. The actomyosin network assembles at the base of the furrow canal and forms a polygonal network in early cellularization. This transforms into a contractile ring in mid cellularization. Finally, ring constriction partially closes the base of the cell during late cellularization (*Kiehart, 1990*; *Krueger et al., 2019*; *Young et al., 1993*; *Royou et al., 2004*; *Schejter and Wieschaus, 1993a*; *Xue and Sokac, 2016*). The actomyosin ring organization and constriction in cellularization are similar to conventional cytokinesis in somatic cells (*Tram et al., 2002*).

The GTPase exchange factor, RhoGEF2, localizes at the furrow tip in cellularization, and its depletion results in inhibition of ring constriction in cellularization (*Padash Barmchi et al., 2005*; *Grosshans et al., 2005*; *Wenzl et al., 2010*). Optogenetic activation of the catalytic domain of RhoGEF2 in the assembly phase of Myosin II leads to increased Myosin II recruitment, resulting in premature and enhanced constriction (*Krueger et al., 2019*). Moreover, overexpression of RhoGEF2 and activated Rho1 mutant Rho1V14 results in contracted S2 cell morphology (*Rogers et al., 2004*). On the other hand, microinjection of Rho1 inhibitor, C3 exoenzyme, and dominant negative N19Rho alters the actin cytoskeleton and disrupts cellularization (*Crawford et al., 1998*). Rho-GTP activates the kinases Drok and Drak that in turn activate Myosin II via phosphorylation. Further, Rok and Drak mutant embryos and addition of a pharmacological inhibitor of Rok, Y-27632, show impaired actomyosin contractility due to reduced Myosin II phosphorylation (*Chougule et al., 2016*; *Krajcovic and Minden, 2012*; *Xue and Sokac, 2016*). Phospho-deficient mutants of the light chain of Myosin II, Squash, also show loss of contractility during cellularization (*Xue and Sokac, 2016*).

In contrast to RhoGEF2 function, RhoGAP domain-containing proteins are likely to play a key role in inhibiting the activation of Myosin II. The function of RhoGAPs during tissue morphogenesis has been generally less studied as compared to RhoGEFs. Depletion of RhoGAPs, RGA3 and RGA4 in the *Caenorhabditis elegans* embryo leads to enhanced contractility of the anterior cortex due to increased recruitment of Myosin II (*Regev et al., 2017*; *Schmutz et al., 2007*). CGAP, a RhoGAP identified in *Drosophila* embryos, plays a role in inhibiting apical constriction in gastrulation (*Mason et al., 2016*). CGAP mutant embryos also show increased constriction of actomyosin rings in cellularization (*Mason et al., 2016*). The hexagonal phase of actomyosin organization has been shown to be resilient to activation by Myosin II recruitment, and RhoGAPs may function at this stage to inhibit ring constriction (*Krueger et al., 2019*). However, analysis of RhoGAP function in a spatiotemporal manner in regulating actomyosin ring contractility has not been studied in cellularization. Here, we report the characterization of a RhoGAP called **G**TPase **r**egulator **a**ssociated with **f**ocal adhesion kinase (GRAF) in regulating contractile ring formation in cellularization in *Drosophila* embryogenesis.

GRAF is a multidomain protein containing a BAR, PH, RhoGAP and SH3 domain (*Lundmark et al., 2008*). GRAF was originally identified as a binding partner of the C-terminal domain of focal adhesion kinase (*Hildebrand et al., 1996*; *Taylor et al., 1998*). The BAR and PH domains are important for the recruitment of GRAF to the membrane, and RhoGAP domain specifically regulates the GTPase activity of RhoA (*Doherty and Lundmark, 2009*; *Doherty et al., 2011*; *Eberth et al., 2009*; *Holst et al., 2017*; *Lundmark et al., 2008*). GRAF-deficient smooth muscle cells in mice and Xenopus embryo extracts have increased Rho-GTP levels (*Bai et al., 2013*; *Doherty et al., 2011*). GRAF loss leads to depletion of epithelial markers and transition to mesenchymal markers, giving rise to enhanced migratory ability (*Regev et al., 2017*). We assessed the role of GRAF in contractile ring formation in *Drosophila* cellularization. We found that GRAF loss showed a distinct hyper constriction phenotype dependent upon its RhoGTPase activity during cellularization. GRAF was present at the furrow tip in early stages of cellularization during Myosin II assembly, enriched in mid cellularization and depleted from the ring in late stages. *Graf* depletion led to increased Myosin II recruitment at the contractile ring. Loss of RhoGEF2 and Myosin II activation suppressed the hyper constriction phenotype of GRAF. Our studies show a crucial role for GRAF in inhibiting actomyosin contractility in cellularization.

## Results

### GRAF depletion leads to premature constriction in early cellularization and hyper constriction in mid and late cellularization

The GRAF protein from *Drosophila* is a multidomain protein containing a BAR, PH, RhoGAP and SH3 domain (*Figure 1A*) from the N terminus to the C terminus. This domain organization is conserved across the multicellular organisms, *C. elegans*, *Drosophila*, *Xenopus*, chick, zebrafish and humans (*Figure 1—figure supplement 1A, B*; *Contrino et al., 2012*). The closest homologue from *Saccharomyces cerevisiae*, RGA2, contains only a RhoGAP domain. *Schizosaccharomyces pombe* contains a similar protein at the *rga2* locus containing a RhoGAP and PH domain, whereas the *gacJJ* locus in Dictyostelium contains a PH, RhoGAP and SH3 domain.

*Drosophila* cellularization occurs by plasma membrane furrow extension between adjacent nuclei from 3 µm to 40 µm in approximately 45 min at 25°C (*He et al., 2016*). The nuclei are spherical in early stages and elongate during mid cellularization. Contractile ring assembly initiates at the base of the furrow tip with Myosin II recruitment in foci in a polygonal network in early stages of cellularization (*Figure 1B*). Constriction of the actomyosin-based contractile ring occurs during mid and late cellularization at the base of elongated nuclei. To assess the function of the GRAF, RNAi knockdown and Crispr Cas-9 strategy were used to generate embryos depleted of GRAF protein.

*Graf* shRNA1 (*Graf$^i$*) (*Figure 1C*) and *Graf* shRNA2 (*Graf$^{2i}$*) (*Figure 1—figure supplement 2A*) were crossed to *maternal*-Gal4 lines to express the shRNA during oogenesis and embryogenesis. F1 females containing the Gal4 and shRNA gave embryos that were lethal at 24 hr (*Graf$^i$* 94% were lethal, n = 532 embryos, and *Graf$^{2i}$* 98% were lethal, n = 540 embryos). Control and mutant embryos were stained with fluorescently coupled phalloidin, which marks cortical F-actin (*Figure 1C, D*). During early stages of cellularization (furrow length 3–6 µm), the F-actin network at the basal part of furrow was organized as a polygonal array (white arrowheads, *Figure 1C*; *He et al., 2016*; *Xue and Sokac, 2016*). The polygonal architecture transitioned to a circular shape on constriction at the furrow tip in mid cellularization (furrow length 6–16 µm) and constricted further in late cellularization (furrow length 16–40 µm) (Figure 1B; *Xue and Sokac, 2016*). *Graf$^i$* and *Graf$^{2i}$* expressing embryos did not show the polygonal F-actin organization in early stages (*Figure 1C*, *Figure 1—figure supplement 2A*). In *Graf$^i$* embryos, the furrow tips contained wavy edges with a loss of contact between adjacent furrow tips (white arrowheads, *Figure 1C*). There was enhanced constriction in mid and late stages of cellularization (*Figure 1C*). The nuclei had a bottleneck appearance in *Graf$^i$* embryos possibly due to premature constriction and squeezing of nuclei in mid cellularization (*Figure 1D*). Quantification of the area of the contractile rings showed a significant decrease in *Graf$^i$* as compared to controls in early, mid and late cellularization (*Figure 1E*).

In addition, loss-of-function mutants at the *Graf* locus were generated by the Crispr-Cas9 strategy. The transgenic fly stock containing a guide RNA (gRNA) against exon 1 (at 2751 residue from transcription start site) of *Graf* was crossed to maternally expressing Cas9 (*nanos*-Cas9) (*Ren et al., 2013*; *Zirin et al., 2020*). A single gRNA against exon 1 of *Graf* was used to avoid perturbing another putative gene CG8960 present in the intronic region of *Graf* in the reverse direction. Embryos obtained from F1 females expressing the gRNA and Cas9 labeled with fluorescently coupled phalloidin gave ring hyper constriction similar to *Graf$^i$* and *Graf$^{2i}$* (*Figure 1—figure supplement 2A*). 155 independent lines from this cross were propagated and screened for crosses that yielded reduced progeny of putative homozygous mutant flies (*Figure 1—figure supplement 2B*, Materials and methods). PCR amplification of exon 1 followed by DNA sequencing was carried out from homozygous flies from 18 lines, which gave reduced progeny, and 3 lines that were similar to controls. The 18 lines with reduced progeny numbers showed small insertions or deletions, leading to a frameshift, thereby causing a premature stop codon in the *Graf* exon 1 region. The three fly lines that served as healthy controls also showed small insertions or deletions but the coding frame was intact (*Figure 1—figure supplement 2C*). The fly line CR57, named *Graf$^{CR57}$* henceforth, contained a stop codon at the 27th amino acid and was chosen for further analysis (*Figure 1A*). *Graf$^{CR57}$*/*Graf$^{CR57}$* homozygous adult females gave embryos with 71% lethality at 24 hr (n = 336). Like the *Graf$^i$* expressing embryos, *Graf$^{CR57}$* embryos obtained from homozygous *Graf$^{CR57}$*/*Graf$^{CR57}$* parents did not show polygonal F-actin architecture in early cellularization. They showed premature ring formation in early cellularization and enhanced ring constriction in mid and late cellularization (white arrowheads, *Figure 1C*). Quantification of ring area of *Graf$^{CR57}$* embryos in cellularization showed a

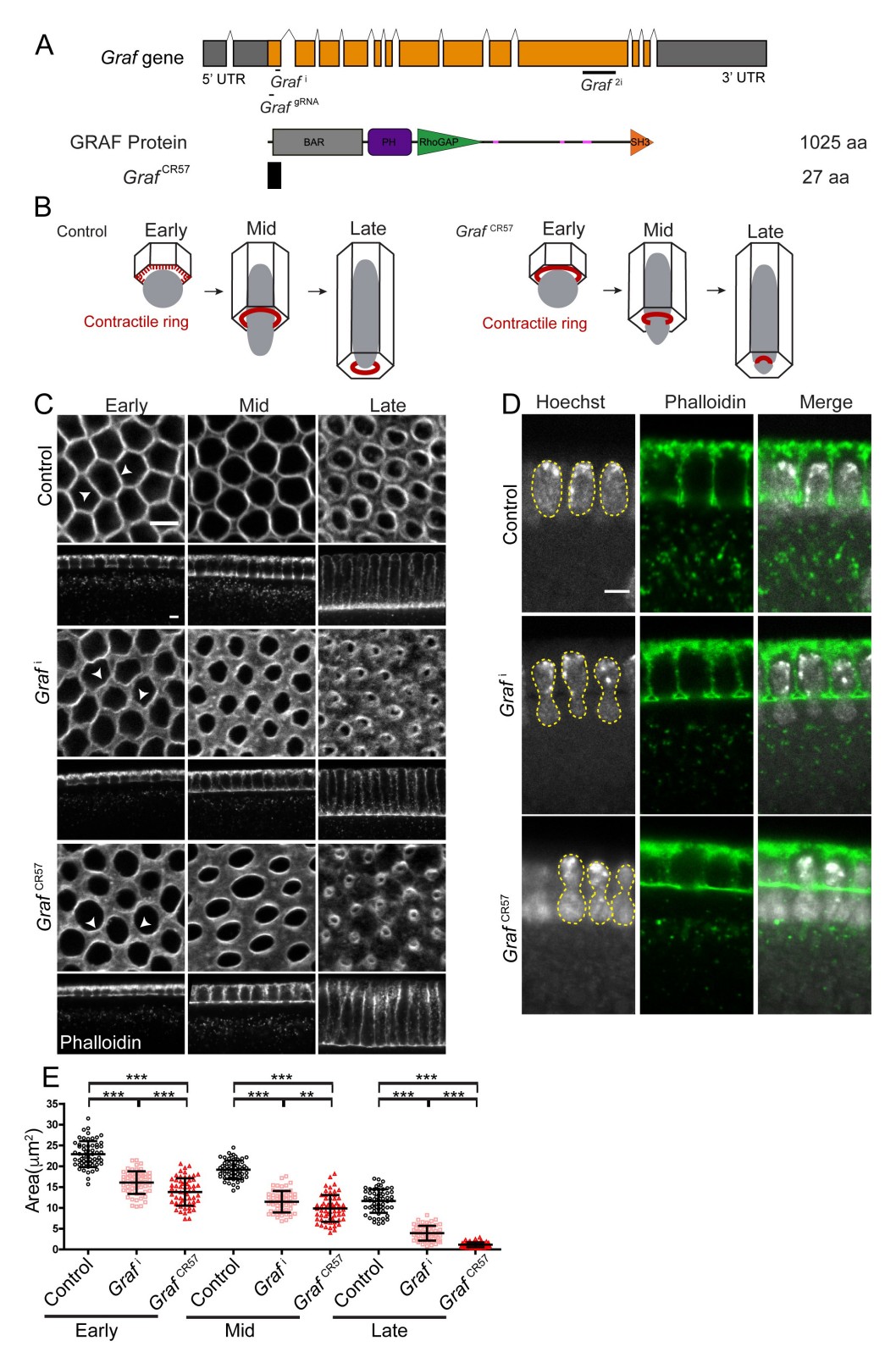

**Figure 1.** GRAF depletion leads to hyper constriction of contractile rings during cellularization. (**A**) GRAF genomic locus is shown with UTR, introns and exons. *Graf*[i], *Graf*[gRNA] target the exon 1 region, and *Graf*[2i] targets the exon 10 region. GRAF protein (1025aa) contains a BAR, PH, RhoGAP and SH3 domain. *Graf*[CR57] has a stop codon at amino acid 28 and is predicted to form a 27aa peptide. (**B**) Schematic depiction of cellularization in the *Drosophila* embryo with plasma membrane (black), nuclei (gray) and contractile ring (red) organization at the base of the furrow. The base of the furrow

*Figure 1 continued on next page*

*Figure 1 continued*

where the contractile ring assembles is hexagonal at the early stage, circular at the mid stage and constricted at the late stage. *Graf*<sup>CR57</sup> mutant embryos show premature ring formation in the early stage and hyper constriction in mid and late stages with nuclei getting squeezed to show a bottleneck phenotype. (C–E) GRAF depletion leads to hyper constricted contractile rings in cellularization. (C) Phalloidin (gray)-labeled furrow tip sections show polygonal organization (white arrowhead) in control embryos, *nanos*-Gal4; *Graf*<sup>i</sup> shows circular organization, and *Graf*<sup>CR57</sup> shows ring constriction (white arrowhead) in the early cellularization. Control embryos show circular rings, and *Graf*<sup>i</sup> and *Graf*<sup>CR57</sup> embryos show constricted rings in mid cellularization. Control embryos show constricted rings, and *Graf*<sup>i</sup> and *Graf*<sup>CR57</sup> embryos show hyper constricted rings in late cellularization (C). *Graf*<sup>i</sup>, 95.7% (n = 70 embryos), and *Graf*<sup>CR57</sup>, 96.8% (n = 31 embryos), show enhanced constriction as compared to controls (n = 54 embryos) in different stages of cellularization. (D) DNA labeled by Hoechst shows the bottleneck appearance during mid cellularization in *Graf*<sup>i</sup> (48.1% show bottleneck nuclei, n = 27 embryos in different stages of cellularization) and *Graf*<sup>CR57</sup> (52.94% show bottleneck nuclei, n = 34 embryos in different stages of cellularization) (yellow line marks nuclei morphology). (E) Scatter plot shows area of the ring in the control, *Graf*<sup>i</sup> and *Graf*<sup>CR57</sup> during early (3–6 μm furrow length), mid (6–16 μm) and late (16–40 μm) stages of cellularization (n = 60 rings, 10 per embryo, 6 embryos each). Data is represented as mean ± s.d. **p<0.01, ***p<0.001, two-tailed Mann–Whitney test. Scale bars: 5 μm.

The online version of this article includes the following source data and figure supplement(s) for figure 1:

**Source data 1.** Plotted values for ring area analysis.
**Figure supplement 1.** Phylogenetic tree of GRAF across various species.
**Figure supplement 2.** GRAF knockdown phenotypes and the Crispr-Cas9 strategy for generating a GRAF null mutant.

---

significant decrease in early, mid and late stages as compared to controls. The ring constriction defects were enhanced in *Graf*<sup>CR57</sup> as compared to *Graf*<sup>i</sup> embryos (*Figure 1E*). *Graf*<sup>CR57</sup> embryos also had bottleneck-shaped nuclei in cellularization (*Figure 1D*). Loss of polygonal F-actin organization and premature ring constriction at the furrow tip is also seen in embryos depleted of Bottleneck, a phospholipid-binding and actin-organizing protein, in cellularization (*Krueger et al., 2019*; *Reversi et al., 2014*; *Schejter and Wieschaus, 1993b*). Taken together, GRAF-depleted embryos showed premature and untimely constriction of the F-actin network at the furrow tip in cellularization, thereby giving rise to bottleneck-shaped nuclei. GRAF function is therefore likely to be involved in inhibiting ring constriction in cellularization (*Figure 1B*).

## GRAF is enriched at the contractile ring during mid cellularization along with Myosin II

In order to visualize the distribution of GRAF in *Drosophila* embryos during cellularization, we generated an antibody against full-length GRAF protein (see Materials and methods). Control embryos were co-stained with GRAF and the polarity protein Dlg to mark lateral membranes. GRAF antibody stained the furrow tip basal to Dlg throughout cellularization (yellow arrowhead, sagittal section, *Figure 2A*). The grazing section across the furrow tip showed an enrichment of GRAF recruitment at edges (white arrowhead, *Figure 2A*) and a reduction at vertices in early cellularization (yellow arrowhead, *Figure 2A*). GRAF antibody staining increased at regions corresponding to tight edges in the Dlg staining during mid cellularization and was occasionally absent from curved regions (yellow arrowhead, *Figure 2A*). GRAF antibody staining was weakly present in rings in late cellularization (*Figure 2A, B*). The GRAF staining was lost in 100% *Graf*<sup>CR57</sup> embryos as compared to controls (*Figure 2A*). Quantification of the cortex to cytosol fluorescence showed a significant reduction in intensity as compared to controls (*Figure 2C*). *Graf*<sup>i</sup> expressing embryos showed a depletion in GRAF antibody staining in 64% of the embryos, and the cortex to cytosol intensity was significantly decreased as compared to controls (*Figure 2—figure supplement 1A, B*). The greater loss of GRAF in *Graf*<sup>CR57</sup> also correlated with more severe defects in ring constriction seen in *Graf*<sup>CR57</sup> as compared to *Graf*<sup>i</sup> (*Figure 1E*). *Graf*<sup>CR57</sup> therefore led to a complete loss of function, and *Graf*<sup>i</sup> led to a partial loss-of-function phenotype for *Graf*.

To visualize GRAF recruitment dynamics in cellularization in living embryos, we generated a C terminally tagged GRAF-GFP in the UASp vector. We expressed the GRAF-GFP transgene with *maternal-Gal4* along with fluorescently labeled Myosin II light chain, Squash (Sqh), *sqh*; Sqh-mCherry (Sqh-mCherry) (*Videos 1* and *2*). The GRAF-GFP and Sqh-mCherry fluorescence was visualized in cellularization, and relative intensities in a single optical plane were quantified with a line segment passing through edges across adjacent furrow tips (yellow line, *Figure 2D, E*). GRAF-GFP was already present at edges in early cellularization during the Myosin II assembly phase when Sqh-mCherry was seen in foci. GRAF-GFP increased further along with Sqh-mCherry at the start of mid cellularization.

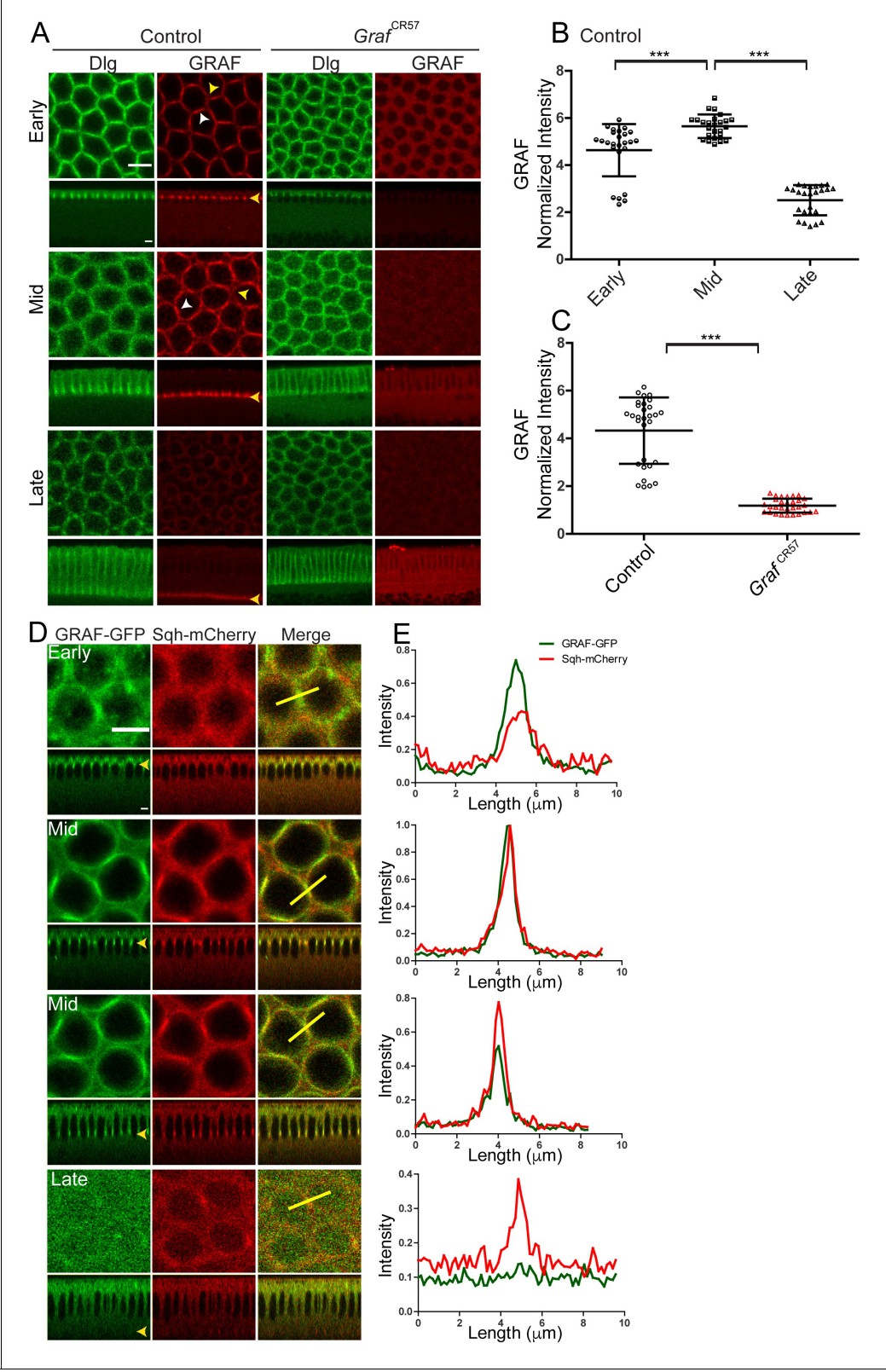

**Figure 2.** GRAF protein is enriched at the furrow in mid cellularization. (**A–C**) GRAF localizes at the furrow tips in cellularization. (**A**) Control embryos (n = 36 embryos) immunostained with GRAF (red) and Dlg (green) show GRAF enrichment at the furrow tip in early and mid cellularization (white arrowhead shows edge enrichment, whereas yellow arrowhead shows curved region with weaker signal). GRAF is decreased from the furrow tip in late cellularization. *Graf*^CR57^ (100%, n = 11) embryos show depletion of GRAF antibody in cellularization. (**B**) Quantification of GRAF antibody fluorescence

*Figure 2 continued on next page*

*Figure 2 continued*

intensity as cortex to cytosol ratio shows increased staining in mid cellularization as compared to early and late cellularization (n = 25 rings, 5 per embryo, 5 embryos per stage, 25 embryos) (furrow length used for early = 4–6 μm, mid = 7–11 μm, late = 17–31 μm). (C) $Graf^{CR57}$ (furrow length range: 3–24 μm) shows loss of GRAF antibody staining intensity compared to controls (furrow length range: 5–25 μm) (n = 30 rings, 5 per embryo, 6 embryos). Data is represented as mean ± s.d. ***p<0.001, two-tailed Mann–Whitney test. (D, E) GRAF-GFP (D, green) colocalizes with Sqh-mCherry (D, red) in early and mid cellularization, whereas GRAF-GFP is cytosolic in late cellularization. Yellow arrowhead in sagittal images shows furrow tip enrichment in mid cellularization, whereas it reduces at the tip in late cellularization. These are representative images from one of n = 3 embryos. (E) The yellow bar depicted in the merged image is used for quantification shown in the plots. Scale bars: 5 μm.

The online version of this article includes the following source data and figure supplement(s) for figure 2:

**Source data 1.** Plotted values for fluorescence intensity analysis.
**Figure supplement 1.** GRAF knockdown shows reduction in GRAF antibody staining in cellularization.
**Figure supplement 1—source data 1.** Plotted values for fluorescence intensity analysis.

GRAF-GFP was lost from the furrow in late cellularization when Sqh-mCherry was present in the constricted ring (*Figure 2D, E*, *Videos 1* and *2*).

In summary, GRAF is present at edges in the furrow tip in early cellularization when Rho-GTP along with Myosin II are beginning to assemble and are present in a polygon. Bottleneck, a developmentally regulated protein present in the *Drosophila* blastoderm embryo, has been previously shown to have a similar distribution with enrichment at the furrow tip during the polygonal phase along with a role in inhibition of constriction (*Schejter and Wieschaus, 1993a*; *Schejter and Wieschaus, 1993b*). The spatial and temporal changes in distribution of GRAF correlate with the dynamic restructuring of the actomyosin network in cellularization.

## GRAF depletion leads to enhanced contractility of the basal actomyosin network

We performed laser ablations to assess contractility of the basal actomyosin network in $Graf^{CR57}$. Sqh-mCherry was used to mark the furrow tip for laser ablations in controls and $Graf^{CR57}$. Laser ablations were performed in a line cutting across contractile rings at the furrow tip in early and mid stages of cellularization (yellow dashed line, *Figure 3A, E*, *Video 3*). The ablation across the ring caused a break in the actomyosin network and led to movement of the ablated edges. The rate of displacement of these ablated edges is dependent upon the tension in the actomyosin network. The rate of movement and the final displacement of the ablated edges at approximately 70 s were calculated for controls and $Graf^{CR57}$. We found a significant increase in rate of displacement of the edges of the contractile rings in $Graf^{CR57}$ embryos as compared to controls in early and late stages (*Figure 3B, F*). $Graf^{CR57}$ embryos showed an approximately fourfold increase in maximum displacement at 70 s as compared to controls in both early and mid cellularization (*Figure 3C, G*). Similarly, initial recoil velocities of the edges after ablations were

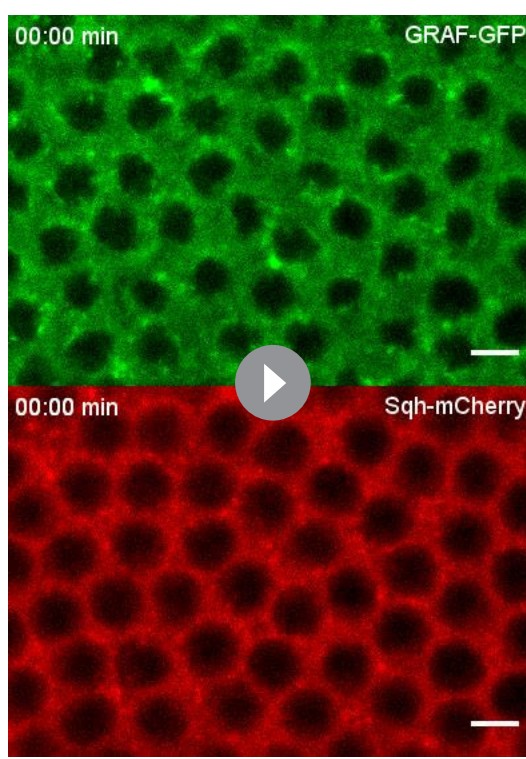

**Video 1.** GRAF-GFP and Sqh-mCherry distribution during ring constriction in cellularization. The video shows living embryos expressing GRAF-GFP (green, above) and Sqh-mCherry (red, below) during cellularization. There is enrichment of GRAF-GFP and Sqh-mCherry at the furrow in mid cellularization. GRAF-GFP becomes cytoplasmic, and Sqh-mCherry remains during late cellularization. Sum projections of five stacks are shown at each time frame of the video. Scale bar = 5 μm.
https://elifesciences.org/articles/63535#video1

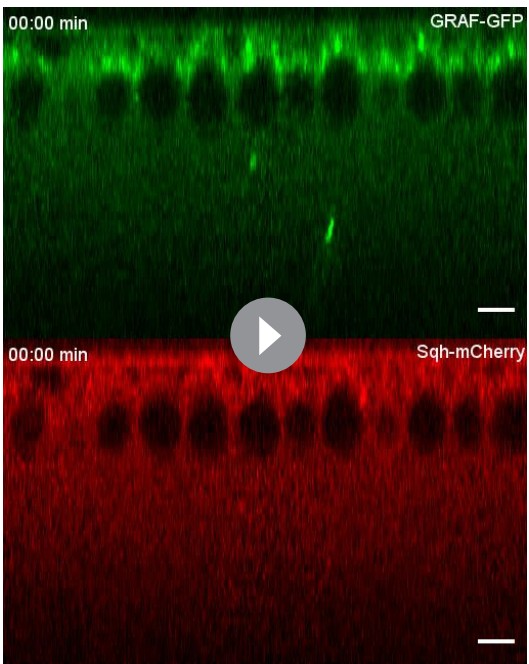

**Video 2.** GRAF-GFP and Sqh-mCherry distribution on the furrow in sagittal sections. Sagittal sections were extracted to show distribution of GRAF-GFP (green, above) and Sqh-mCherry (red, below) during cellularization. Note the loss of GRAF-GFP in late cellularization, whereas Sqh-mCherry is still visible at the furrow tip (white arrows at the 42.26 min time point). GRAF-GFP also labels tube-like structures beneath the furrow in early and mid cellularization. Scale bar = 5 μm.

https://elifesciences.org/articles/63535#video2

increased by approximately twofold in *Graf*[CR57] embryos in early and mid stages of cellularization as compared to controls (*Figure 3D, H*). This increased displacement and recoil velocity of the ring edges suggest that the actomyosin network in *Graf*[CR57] has increased contractility. The increased contractility in *Graf* mutants in early and mid stages, along with the distinct recruitment of GRAF in early and mid stages (*Figure 2*), shows that GRAF plays a crucial role in resisting Myosin II-dependent contractility during early and mid stages of cellularization.

## GRAF RhoGAP domain restricts contractile ring constriction during cellularization

Since GRAF has a RhoGAP domain, we assessed the recruitment dynamics of Rho-GTP at the contractile rings in the furrow tip in *Graf*[CR57]. Rho-GTP was visualized by GFP-tagged Rho-GTP-binding domain (RBD) of Anillin (AnillinRBD-GFP) expressed under the control of *ubiquitin* promoter (*Mason et al., 2016*; *Munjal et al., 2015*) along with Sqh-mCherry. AnillinRBD-GFP was present as foci that colocalized with Sqh-mCherry in early cellularization during the assembly of the actomyosin complex at the furrow tip. AnillinRBD-GFP was present along with Sqh-mCherry at the contractile ring in mid and late cellularization in images (*Figure 4A*, *Video 4*). Line scans across the edges of adjacent rings in furrow tips were used from single optical planes to estimate the fluorescence intensity relative to the maximum seen in cellularization in each fluorescent tag (yellow line,

*Figure 4B*). Anillin-RBD fluorescence increased from early to mid stages and decreased in late stages of cellularization in control embryos (*Figure 4B*). Even though the fluorescence was reduced in late stages, AnillinRBD-GFP was present at the furrow tip along with Sqh-mCherry (*Figure 4A, B*) when GRAF-GFP was lost in late cellularization (*Figure 2*). AnillinRBD-GFP and Sqh-mCherry signal was more spread during early and mid stages of cellularization in *Graf*[CR57] embryos compared to sharp colocalization peaks in control (*Figure 4C, D*, *Video 4*). This increase in Rho-GTP in early and mid stages of cellularization correlated with hyper contractility in *Graf*[CR57] embryos.

We further analyzed the role of the RhoGTPase domain of GRAF in regulating ring constriction during cellularization. We generated a fluorescently tagged *Graf* transgene containing a deletion of the Rho-GAP domain, UASp-GRAFΔRhoGAP-GFP (*Figure 4E*). We expressed GRAFΔRhoGAP-GFP and GRAF-GFP maternally in the *Graf*[CR57] mutant background to assess the effect of the Rho-GAP deletion on actomyosin ring constriction in cellularization. Live imaging of embryos of the *Graf*[CR57]; GRAF-GFP and *Graf*[CR57]; GRAFΔRhoGAP-GFP genotypes was carried out with Sqh-mCherry (*Figure 4F*). Similar to GRAF-GFP, GRAFΔRhoGAP-GFP recruited to the furrow until mid cellularization and was lost from the furrow in late cellularization similar to GRAF-GFP (*Figure 4—figure supplement 1A, B*). However, GRAFΔRhoGAP-GFP appeared in punctae in early cellularization, leading to higher relative fluorescence as compared to GRAF-GFP in early cellularization (*Figure 4—figure supplement 1A, B*). The ring area in *Graf*[CR57] embryos was significantly lower than controls during cellularization (*Figure 4F, G*). This hyper constriction phenotype in *Graf*[CR57] was suppressed in *Graf*[CR57]; GRAF-GFP embryos. *Graf*[CR57]; GRAFΔRhoGAP-GFP embryos, on the other hand, showed hyper constriction of rings similar to *Graf*[CR57] (*Figure 4F, G*, *Videos 5* and *6*). Thus, the full-length

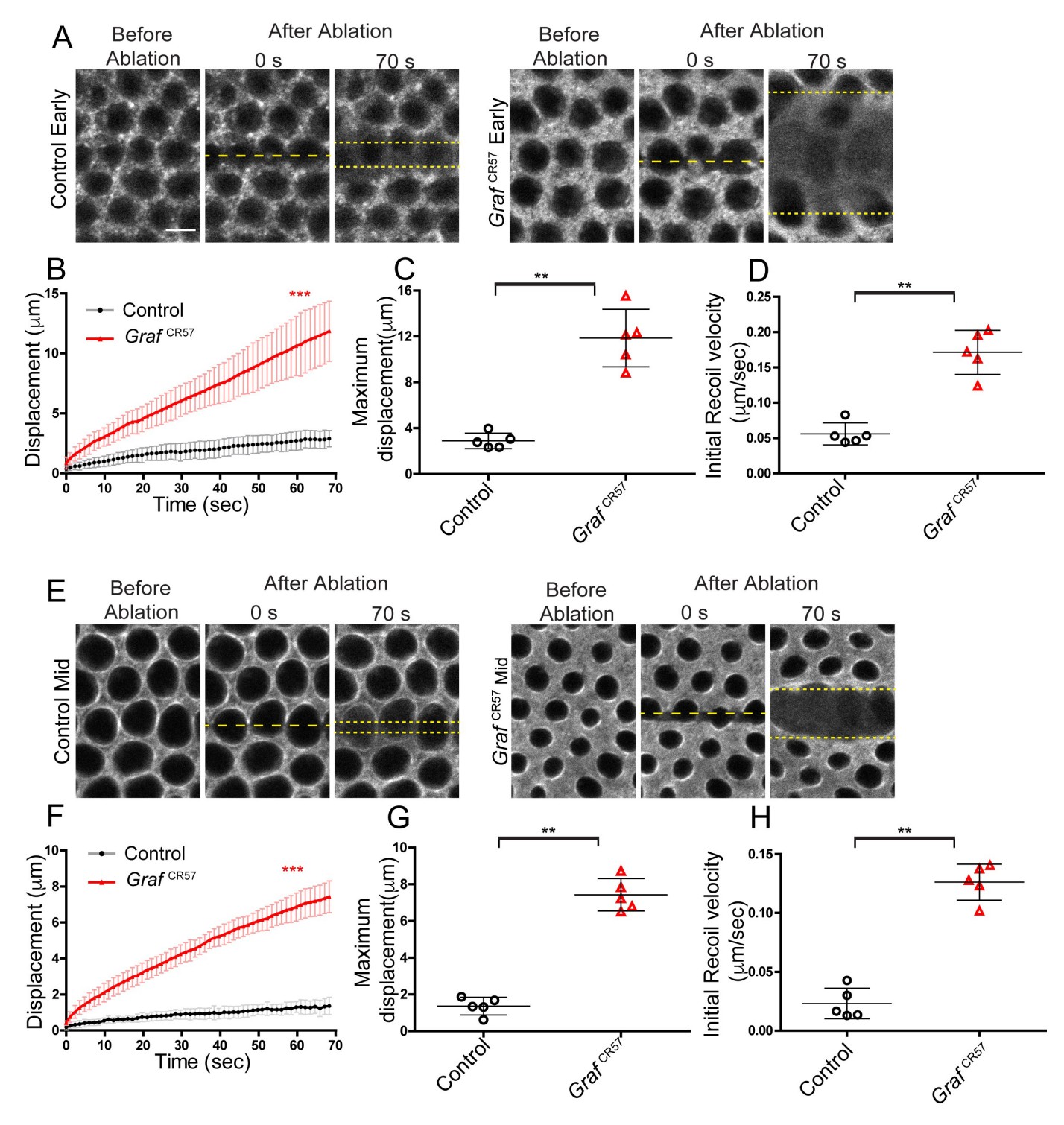

**Figure 3.** *Graf* mutant embryos show hyper contractility during early and mid cellularization. Laser ablation was performed using line at the furrow tip during early and mid cellularization. (**A**, **E**) Representative images show control and *Graf*CR57 mutant embryos with the region before, at 0 s (marked with yellow dotted line) and 70 s after ablations (recoil region marked with yellow dotted line) in early and mid cellularization, respectively. (**B**, **F**) Quantifications of ring displacement after laser ablation during early and mid phase (control:black line, n = 5 embryos; *Graf*CR57 mutant:red line, n = 5 embryos each). (**C,D,G,H**) Scatter plot showing maximum displacement (**C**, **G**) of ring and initial recoil velocity (**D**, **H**) after laser ablation during early and mid phase (n = 5 embryos each). Data is represented as mean ± s.d.**p<0.01, ***p<0.001, two-tailed Mann–Whitney test.

The online version of this article includes the following source data for figure 3:

**Source data 1.** Plotted values for displacement and initial recoil velocity.

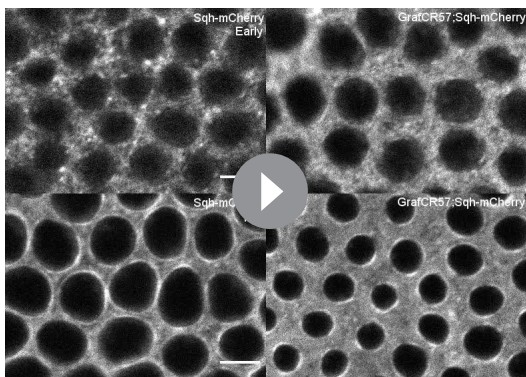

**Video 3.** *Graf*<sup>CR57</sup> embryos show contractile ring recoil upon ablations during cellularization. Live imaging of Sqh-mCherry control (left) and *Graf*<sup>CR57</sup> (right) containing Sqh-mCherry (gray) subjected to laser ablations (time = 0 s) at the regions marked in a yellow line during early (above) and mid (below) cellularization. *Graf*<sup>CR57</sup> shows higher ring recoil as compared to controls. Scale bar = 5 μm.
https://elifesciences.org/articles/63535#video3

GRAF-GFP suppressed the constriction defect of *Graf*<sup>CR57</sup>. The RhoGAP domain in GRAF protein was necessary for restricting ring constriction during cellularization.

## GRAF overexpression leads to decrease in contractile ring constriction during cellularization

Overexpression of GRAF was achieved by crossing Graf<sup>EP09461</sup> to *mat*-Gal4 (GRAF-OE). Graf<sup>EP09461</sup> contains a gypsy transposon tagged line with a UASp element inserted in the 5′UTR region of GRAF in the same orientation of native gene (*Figure 5A*). GRAF-OE embryos showed a significant increase in GRAF antibody staining as compared to controls (*Figure 5B, C*).

*Graf* depletion led to hyper constriction of ring area in cellularization (*Figure 4G*). We further assessed if GRAF overexpression led to inhibition of ring constriction in cellularization. GRAF-OE embryos were imaged live in cellularization with Sqh-mCherry (*Video 7*). We found that the ring area was significantly larger in GRAF-OE as compared to controls from early stages of cellularization (*Figure 5D, E*). These data show that GRAF overexpression leads to inhibition of ring constriction during cellularization consistent with increased RhoGTPase activity.

## GRAF depletion and overexpression leads to changes in Myosin II distribution at the contractile ring during cellularization

We assessed the distribution of Myosin II heavy chain, Zipper, in control and *Graf*<sup>CR57</sup> fixed embryos in early, mid and late cellularization by immunostaining. We found that the Zipper fluorescence at the furrow tip relative to the cytoplasm was significantly increased in *Graf*<sup>CR57</sup> throughout cellularization as compared to control embryos (*Figure 6A, B*). Zipper foci in *Graf*<sup>CR57</sup> appeared larger than controls in early cellularization, and Zipper was also enriched in between adjacent rings in mid cellularization (white arrowheads, *Figure 6A*).

We also analyzed embryos containing depletion or overexpression of GRAF for Myosin II light chain, Sqh-mCherry distribution and furrow extension dynamics from movies obtained from living embryos. Since Sqh-mCherry marked the furrow tip, it was also used to estimate furrow lengths during cellularization. The dynamics of furrow length increase in the slow and fast phase in GRAF-OE were similar to controls during cellularization (*Figure 6—figure supplement 1A*; *Figard et al., 2013*; *He et al., 2016*; *Lecuit and Wieschaus, 2000*; *Merrill et al., 1988*; *Royou et al., 2004*; *Warn and Magrath, 1983*). In contrast, the slow phase in *Graf*<sup>CR57</sup> was extended and the fast phase started at a later time point (*Figure 6—figure supplement 1A*). The fluorescence at the ring in the furrow tip was estimated per pixel as a ratio to maximum intensity across cellularization (*Figure 6—figure supplement 1B*; see Materials and methods). The levels of Sqh-mCherry increased from early to mid cellularization and then decreased during late cellularization in controls. GRAF-OE embryos showed a delay in Sqh-mCherry peak as compared to controls (*Figure 6—figure supplement 1B*). *Graf*<sup>CR57</sup>, on the other hand, showed a sustained Sqh-mCherry fluorescence throughout cellularization, and it remained significantly higher at later stages of cellularization (*Figure 6C*, *Figure 6—figure supplement 1B*, *Videos 5* and *7*). GRAF-OE and control embryos showed a similar distribution of Sqh-mCherry in foci in early cellularization followed by a more uniform distribution in mid cellularization. GRAF-OE did not affect the steps of transition of Myosin II assembly from foci to uniform distribution even though there was an overall delay in the peak of Sqh-mCherry fluorescence in cellularization (yellow arrowheads, *Figure 6C*).

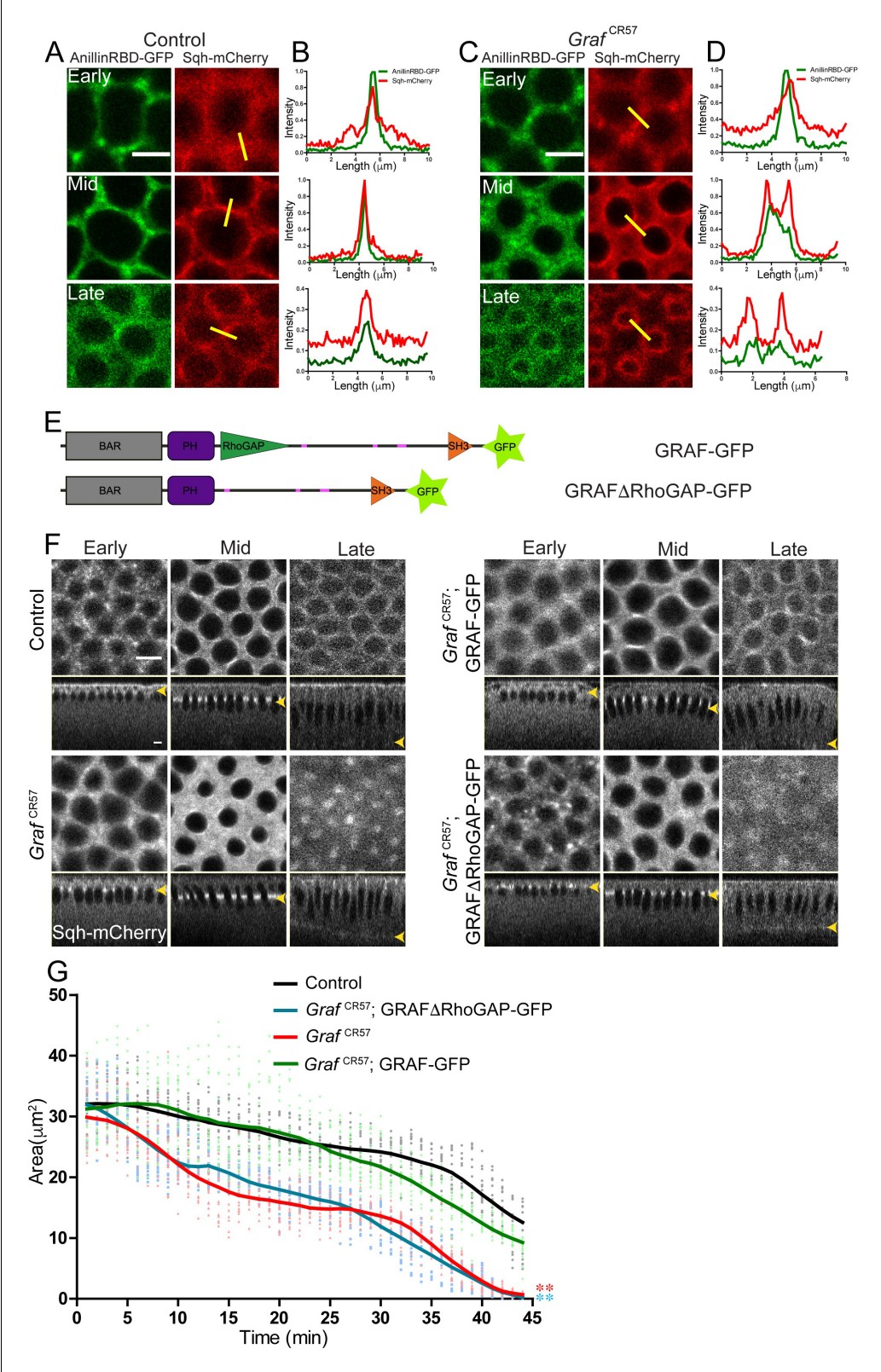

**Figure 4.** RhoGAP domain of GRAF is essential for ring constriction in cellularization. (A–D) AnillinRBD-GFP (A, C, green) colocalizes with Sqh-mCherry (red) in early, mid and late cellularization in controls. In *Graf*<sup>CR57</sup>; AnillinRBD-GFP, the signal is more spread as compared to in between adjacent rings. Representative images from one of n = 3 embryos of controls and *Graf*<sup>CR57</sup> are shown. (B, D) A yellow bar depicted in the Sqh-mCherry image is used for the estimation of the intensity line profile. (E) Schematic showing full-length GRAF protein with GFP and GRAFΔRhoGAP-GFP proteins. (F) *Graf*<sup>CR57</sup>

*Figure 4 continued on next page*

Figure 4 continued

and *Graf*^CR57;GRAFΔRhoGAP-GFP embryos expressing Sqh-mCherry show hyper constriction compared to controls and *Graf*^CR57; GRAF-GFP (yellow arrowhead highlights furrow tip at all stages). **(G)** Quantification shows a significantly lower ring area for *Graf*^CR57, *Graf*^CR57;GRAFΔRhoGAP-GFP as compared to controls and *Graf*^CR57; GRAF-GFP expressing Sqh-mCherry (n = 15 rings, 5 rings per embryo, 3 embryos). Data is represented as mean ± s.d. ***p<0.001, one-way ANOVA, repeated measure with Dunnett's multiple comparison test, the *Graf*^CR57 and *Graf*^CR57;GRAFΔRhoGAP-GFP mutant curves are statistically different when compared to the controls and *Graf*^CR57;GRAF-GFP. Scale bars: 5 μm.

The online version of this article includes the following source data and figure supplement(s) for figure 4:

**Source data 1.** Plotted values for fluorescence intensity and ring area analysis.

**Figure supplement 1.** *Graf*^CR57;GRAFΔRhoGAP-GFP recruitment dynamics.

**Figure supplement 1—source data 1.** Plotted values for fluorescence intensity analysis.

*Graf*^CR57 showed Sqh-mCherry in foci that were somewhat more spread as compared to controls in early cellularization. *Graf*^CR57 showed enhanced Sqh-mCherry fluorescence in between adjacent rings in mid cellularization (yellow arrowheads, *Figure 6C*). *Graf*^CR57 embryos contained a sustained presence of Sqh-mCherry in rings in late cellularization (yellow region in the graph in *Figure 6—figure supplement 1B*). The Sqh-mCherry intensity per pixel was further estimated from rings relative to the inter ring region from five time points in the yellow region (*Figure 6—figure supplement 1B*) in GRAF-OE, *Graf*^CR57, *Graf*^CR57; GRAFΔRhoGAP-GFP and *Graf*^CR57; GRAF-GFP as a readout of change in Myosin II recruitment (*Figure 6D*). The Sqh-mCherry intensity was indeed increased in *Graf*^CR57 and *Graf*^CR57; GRAFΔRhoGAP-GFP and reduced in GRAF-OE (*Figure 6D*). The phenotype of increased Sqh-mCherry in *Graf*^CR57 embryos was suppressed in *Graf*^CR57; GRAF-GFP (*Figure 6D*). Increased Zipper and Sqh-mCherry levels at the furrow on *Graf* depletion and decreased Sqh-mCherry levels on GRAF-OE during cellularization showed that Myosin II recruitment at the furrow tip is inhibited by GRAF, and this was dependent upon its RhoGAP domain.

## GRAF overexpression-driven decrease in contractile ring constriction is suppressed by overexpression of RhoGEF2

The RhoGTPase domain of GRAF was essential for regulating ring constriction during cellularization. GRAF overexpression showed loss of constriction and delayed enrichment of Myosin II. GRAF over-expression is therefore likely to decrease Rho-GTP and phenocopy loss of RhoGEF2. Optogenetic activation of RhoGEF2 leads to increased constriction in the assembly phase in cellularization (*Krueger et al., 2019*) and phenocopies the *Graf*^CR57 mutant (*Figures 1* and *3*). Our previous analysis has shown that RhoGEF2-OE leads to increase in Myosin II at the cortex in syncytial division cycle (*Dey and Rikhy, 2020*). We overexpressed RhoGEF2 maternally in embryos and imaged them live with Sqh-mCherry in cellularization. We found that similar to *Graf*^CR57 overexpression of RhoGEF2 (RhoGEF2-OE) caused ring hyper constriction as compared to controls in cellularization (*Figure 7A, C*). Sqh-mCherry was found in foci that were more spread in RhoGEF2-OE embryos as compared to controls and similar to *Graf*^CR57 mutant embryos (white arrow, *Figure 6C* and *Figure 7A*).

Since relative levels of GAP and GEF are likely to regulate the rate of contractility (*Mason et al., 2016*) and overexpression of RhoGEF2 and GRAF will have opposing effects on levels of Rho-GTP, we tested if phenotypes of RhoGEF2 overexpression could be inhibited by

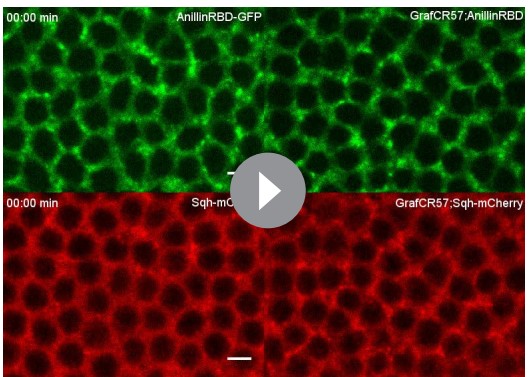

**Video 4.** *Graf*^CR57 embryos show enrichment of AnillinRBD-GFP and Sqh-mCherry distribution during ring constriction in cellularization. The video shows colocalization of AnillinRBD-GFP (green, above, left) and Sqh-mCherry (red, below, left) throughout cellularization. *Graf*^CR57 embryos show enrichment of AnillinRBD-GFP (above, right) and Sqh-mCherry (below, right) between neighboring rings from mid cellularization to late cellularization. Sum projections of five stacks at the furrow tip are used to show each time frame in the video. Scale bar = 5 μm.

https://elifesciences.org/articles/63535#video4

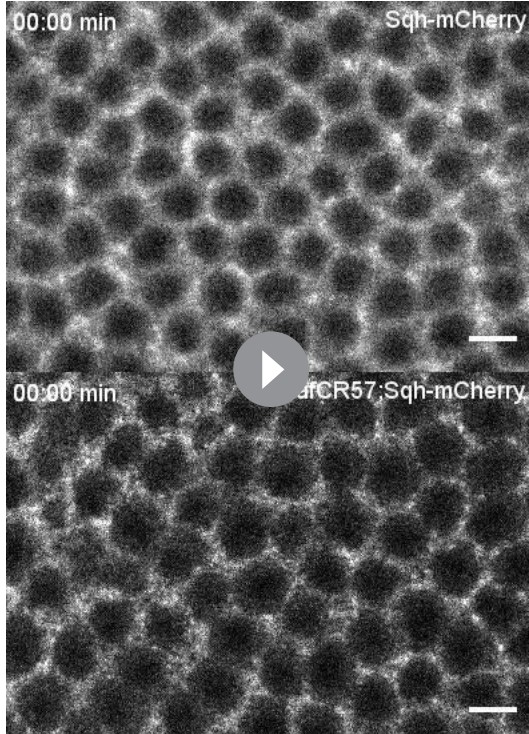

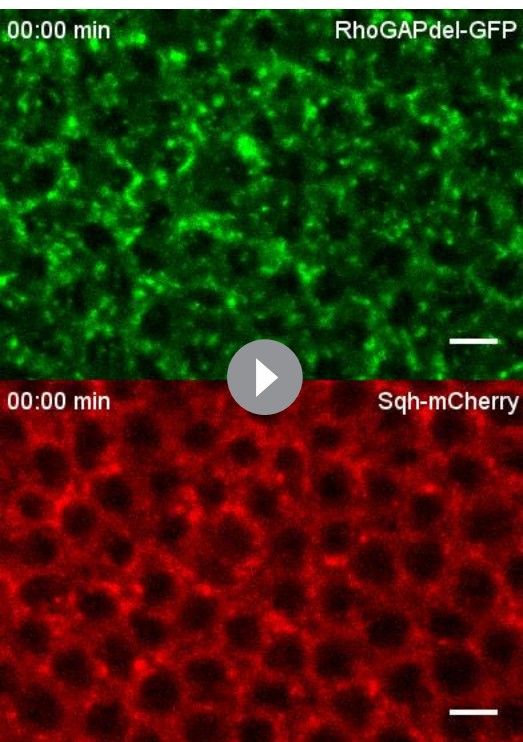

**Video 5.** *Graf*<sup>CR57</sup> embryos expressing Sqh-mCherry show increased ring constriction during cellularization. Live imaging of Sqh-mCherry control (above) and *Graf*<sup>CR57</sup> (below) containing Sqh-mCherry (gray) at the furrow during cellularization. *Graf*<sup>CR57</sup> shows enhanced constriction as compared to controls. Sum projections of five stacks at the furrow tip are shown at each time frame of the video. Scale bar = 5 μm.

https://elifesciences.org/articles/63535#video5

**Video 6.** *Graf*<sup>CR57</sup>; GRAFΔRhoGAP-GFP embryos expressing Sqh-mCherry show enhanced ring constriction during cellularization. Live embryos expressing *Graf*<sup>CR57</sup>; GRAFΔRhoGAP-GFP (green, above) and Sqh-mCherry (red, below) are shown during cellularization. *Graf*<sup>CR57</sup>; GRAFΔRhoGAP-GFP shows enhanced constricted ring compared to controls (*Video 1*). Sum projections of five stacks at the furrow tip are shown at each time frame of the video. Scale bar = 5 μm.

https://elifesciences.org/articles/63535#video6

GRAF overexpression. We found that the ring constriction dynamics in GRAF-OE; RhoGEF2-OE were similar to controls (*Figure 7A, C*). GRAF was recruited to contractile rings in RhoGEF-OE embryos in cellularization (*Figure 7—figure supplement 1A*). However, RhoGEF2-OE embryos showed a decrease in furrow length in cellularization and the rate of furrow extension was lower than controls (*Figure 7—figure supplement 1B*). Late-stage embryos with shorter furrow length as compared controls also showed the presence of GRAF at the furrow tip (*Figure 7—figure supplement 1A*). This implied that the phenotype of enhanced constriction in RhoGEF2-OE was not due to the removal of GRAF. The phenotypes of hyper constriction in RhoGEF2-OE and ring inhibition in GRAF-OE were suppressed in GRAF-OE; RhoGEF2-OE embryos (*Figure 7A, C*; *Video 8*). This suppression was also seen in fixed GRAF-OE; RhoGEF2-OE embryos stained with phalloidin (*Figure 7B*). RhoGEF2-OE embryos showed an increase in Sqh-mCherry at the ring relative to the inter ring in late cellularization as compared to controls. This increase was suppressed in GRAF-OE; RhoGEF2-OE combination (*Figure 7D*). It is possible that the increase in Rho-GTP-Rho-GDP cycling in GRAF-OE; RhoGEF2-OE embryos leads to Rho-GTP levels similar to controls, thereby restoring the rate of constriction.

## GRAF depletion-induced contractile ring hyper constriction in cellularization is suppressed by additional depletion of RhoGEF2

RhoGEF2 depletion inhibits ring constriction during cellularization by decreasing the levels of Rho-GTP (*Padash Barmchi et al., 2005*; *Grosshans et al., 2005*; *Wenzl et al., 2010*). We depleted

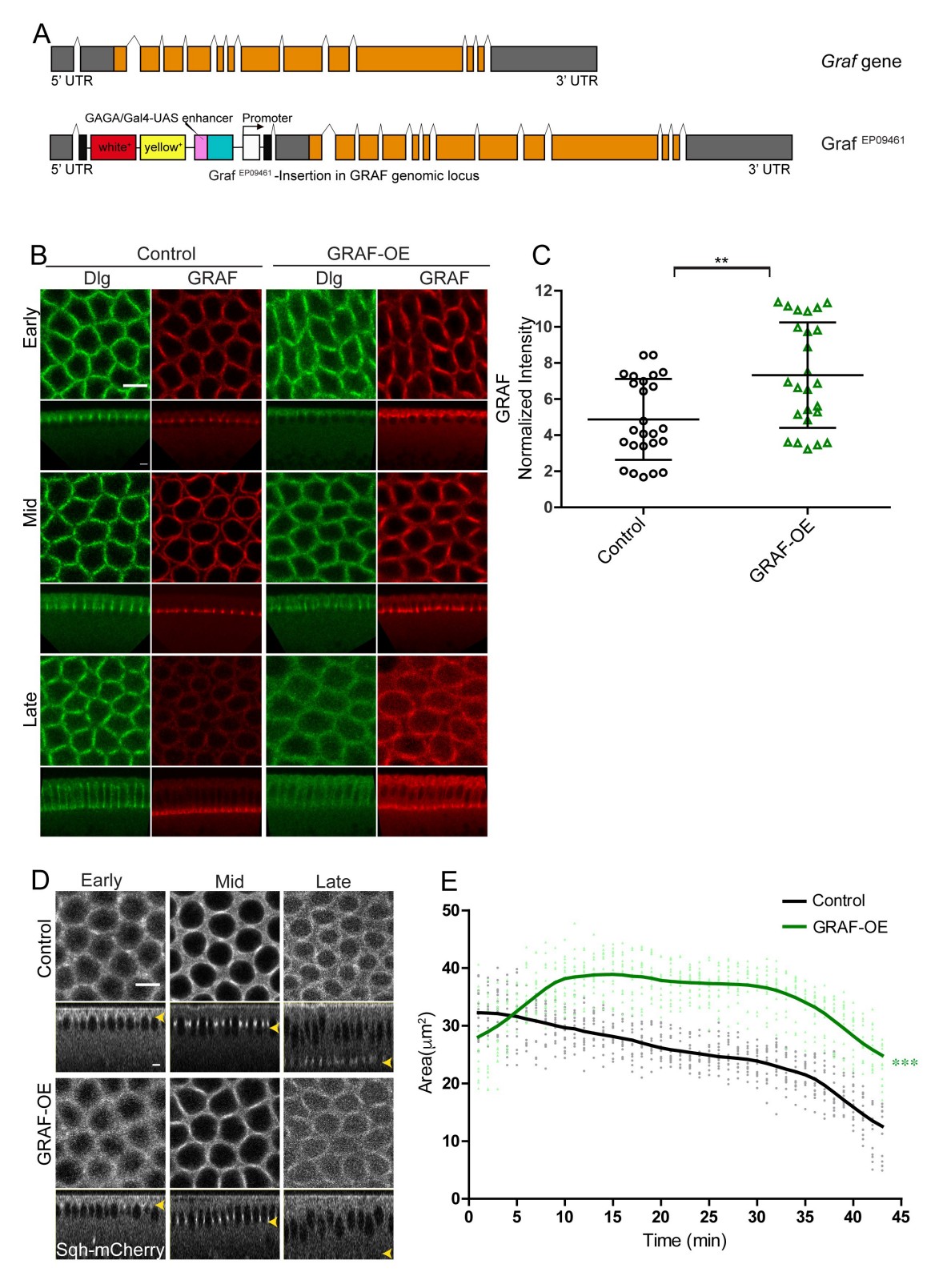

**Figure 5.** GRAF overexpression shows inhibition of ring constriction in cellularization. (**A–C**) Schematic (**A**) shows *Graf* gene and the Graf overexpression line, Graf[EP09461] containing a P-element (with a UAS site) insertion in 5'UTR in an orientation to drive the downstream *Graf* gene. (**B**) GRAF-OE crossed to *mat*-Gal4 shows an increase in GRAF (100%, n = 19) when stained with GRAF (red) and Dlg (green) (control n = 24 embryos). (**C**) Quantification shows an increased GRAF antibody fluorescence in GRAF-OE (furrow length range: 7–19 μm) embryos compared to controls (furrow

*Figure 5 continued on next page*

*Figure 5 continued*

length range: 7–20 µm) (n = 25 rings, 5 per embryo, 5 embryos each). (D) Sqh-mCherry images and (E) area quantification show a significantly higher ring area in GRAF-OE as compared to controls (yellow arrowhead marks the furrow tip) (n = 15 rings, 5 per embryo, 3 embryos)(control values were repeated from *Figure 3* for comparison). Data is represented as mean ± s.d. **p<0.01, ***p<0.001, two-tailed Mann–Whitney test. Scale bars: 5 µm. The online version of this article includes the following source data for figure 5:

**Source data 1.** Plotted values for fluorescence intensity and ring area analysis.

RhoGEF2 by driving maternal expression of RNAi against RhoGEF2 (*RhoGEF2*[i]) and visualized furrow length and ring dynamics with Sqh-mCherry in cellularization (*Kerridge et al., 2016*; *Sherlekar and Rikhy, 2016*). Sqh-mCherry and Zipper were found to be reduced at the furrow in syncytial division cycles in *RhoGEF2*[i] expressing embryos (*Dey and Rikhy, 2020*). *RhoGEF2*[i] expressing embryos showed polygonal furrow tip organization in mid cellularization (*Figure 8A, B, Video 9*), and this phenotype was similar to GRAF overexpression (*Figure 7A, B*). Immunostaining of *RhoGEF2*[i] embryos with GRAF and Dlg antibodies showed that GRAF recruitment occurred at the furrow in early and mid stages of cellularization and was lost in the late stage similar to controls (*Figure 8—figure supplement 1B*). GRAF recruitment therefore did not depend upon RhoGEF2 in cellularization. The furrow extension dynamics in RhoGEF2-depleted embryos were similar to controls with the slow and fast phase of cellularization occurring at the same rate (*Figure 7—figure supplement 1B*). As seen previously for RhoGEF2 mutants, *RhoGEF2*[i] expressing embryos showed loss of constriction in cellularization (*Figure 8A*; *Padash Barmchi et al., 2005*; *Wenzl et al., 2010*). Sqh-mCherry fluorescence appeared diffuse at the furrow tip and was present in a polygonal shape in cellularization in *RhoGEF2*[i] expressing embryos. The *Graf*[CR57]; *RhoGEF2*[i] combination, on the other hand, showed recruitment of Sqh-mCherry and ring constriction similar to control embryos in early and mid cellularization (*Figure 8A, C, Video 9*). *Graf*[CR57]; *RhoGEF2*[i] also rescued the hyper constriction defect of *Graf*[CR57] in late cellularization (35–45 min, *Figure 8C*) even though this was not reversed similar to controls. *RhoGEF2*[i] embryos showed reduced Sqh-mCherry fluorescence in the ring relative to the inter ring region as compared to controls in late cellularization. *Graf*[CR57]; *RhoGEF2*[i] embryos showed a decrease in Sqh-mCherry levels as compared to *Graf*[CR57] even though it remained higher than controls (*Figure 8D*). This increased Sqh-mCherry level also correlated with increased area in late stages (*Figure 8C*). This suppression of hyper constriction of *Graf*[CR57] and lack of constriction in *RhoGEF2*[i] was seen in fixed *Graf*[i]; *RhoGEF2*[i] embryos in cellularization with phalloidin staining (*Figure 8B*). The suppression is likely to occur due to the presence of Rho-GTP levels that give constriction at the same rate as control embryos. This suggested that Rho-GTP was indeed present in *RhoGEF2*-depleted embryos. It is also possible that RhoGEF2 inhibition was incomplete in *RhoGEF2*[i] embryos or another RhoGEF function took over constriction in the absence of RhoGEF2.

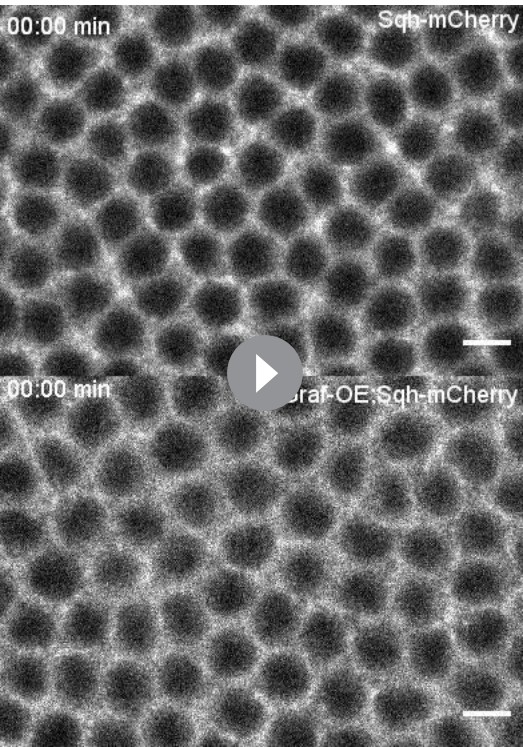

**Video 7.** GRAF-OE embryos expressing Sqh-mCherry show inhibition of ring constriction during cellularization. Live imaging of Sqh-mCherry control (above) and Graf-OE (below) embryo expressing Sqh-mCherry (gray) at the furrow is shown. Graf-OE leads to inhibition of ring constriction when compared to controls. Sum projections of five stacks at the furrow tip are shown at each time frame of the video. Scale bar = 5 µm.

https://elifesciences.org/articles/63535#video7

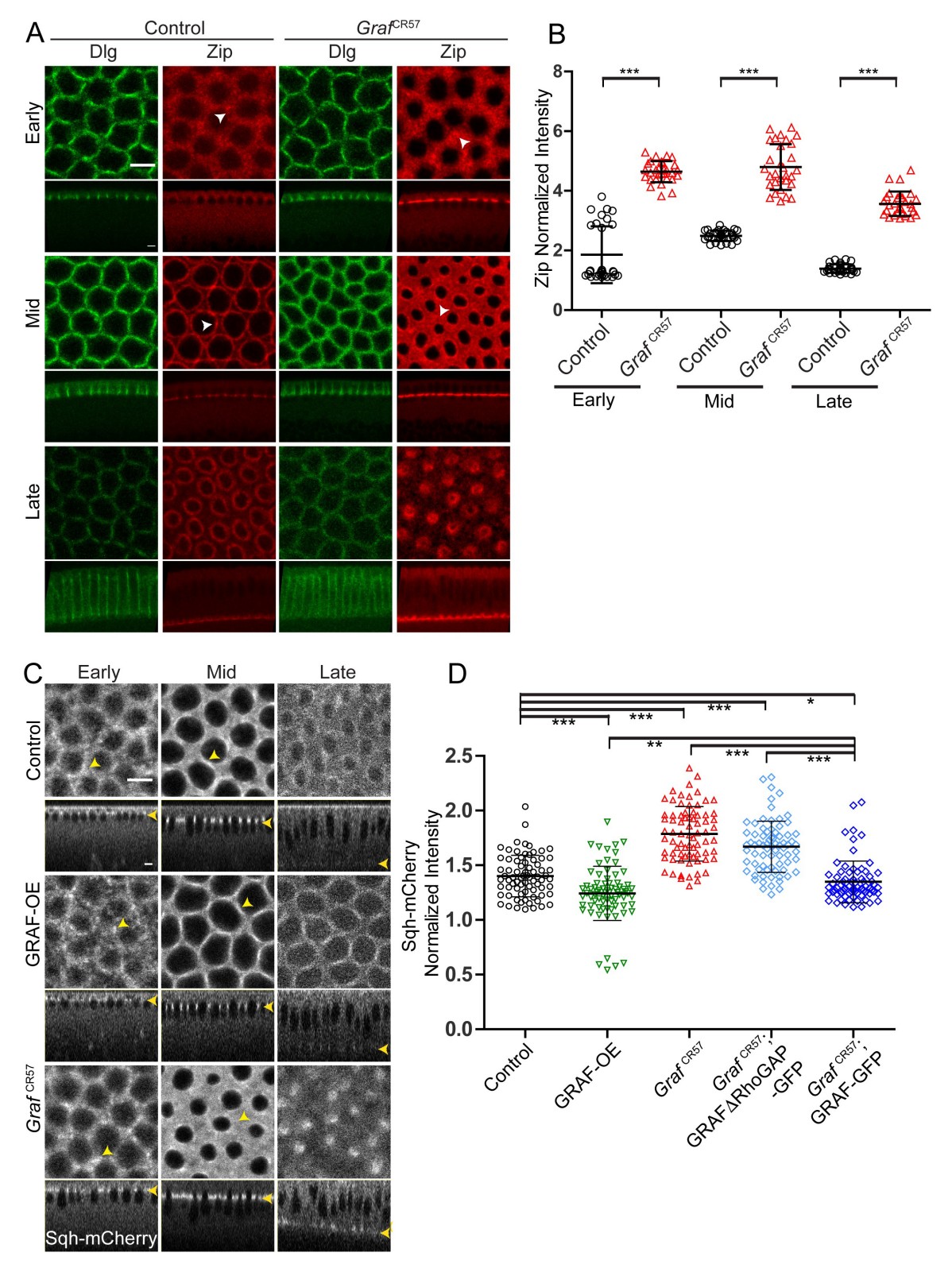

**Figure 6.** *Graf*-depleted embryos show increased Myosin II in cellularization, whereas GRAF overexpression shows reduced Myosin II. (A–D) Fixed images (A) of control and *Graf*CR57 embryos (100%, n = 31 embryos) stained with Dlg (green) and Zipper (red) showing higher Zipper intensity in early (white arrowhead shows Zipper foci), mid (white arrowhead shows Zipper enrichment in control and spreading in *Graf*CR57 mutant) and late cellularization (control n = 16 embryos). (B) Cortex to cytosol ratio of Zipper antibody fluorescence shows a significant increase in *Graf*CR57 as compared

*Figure 6 continued on next page*

*Figure 6 continued*

to controls (n = 30 rings, 10 per embryo, 3 embryos each). (C) Control, GRAF-OE and *Graf*^CR57 embryos expressing Sqh-mCherry (gray) are shown in early (yellow arrowhead marks Sqh-mCherry foci), mid (yellow arrowhead marks Sqh-mCherry enrichment) and late cellularization (yellow arrowhead in sagittal sections marks the furrow tip). (D) Quantification of Sqh-mCherry in late cellularization (yellow region marked in *Figure 6—figure supplement 1B*) in GRAF-OE shows reduction in intensity, whereas *Graf* ^CR57 and *Graf* ^CR57;GRAFΔRhoGAP-GFP shows higher intensity as compared to controls. *Graf* ^CR57; GRAF-GFP shows a rescue in Sqh-mCherry as compared *Graf* ^CR57 (n = 75 rings, 5 rings per time point, 5 time points, 3 embryos each). Data is represented as mean ± s.d. *p<0.05, **p<0.01, ***p<0.001, two-tailed Mann–Whitney test. Scale bars: 5 μm.

The online version of this article includes the following source data and figure supplement(s) for figure 6:

**Source data 1.** Plotted values for fluorescence intensity analysis.
**Figure supplement 1.** Sqh-mCherry recruitment and furrow length analysis in *Graf*^CR57 and GRAF-OE.
**Figure supplement 1—source data 1.** Plotted values for furrow length and fluorescence intensity analysis.

## GRAF depletion-induced contractile ring hyper constriction in cellularization is lost by additional depletion of ROK

Myosin II levels were increased at the furrow tip in *Graf*^CR57 mutant embryos (*Figure 5*). Rho kinase (dRok) activates Myosin II by phosphorylation in a Rho-GTP-dependent manner (*Chougule et al., 2016*; *Xue and Sokac, 2016*). In order to test if activated Myosin II was responsible for hyper constriction, we depleted Rho kinase in *Graf*^CR57 embryos. Rho kinase was depleted by maternal expression of RNAi against Rok (*rok*^i) (*Dey and Rikhy, 2020*; *Zhang et al., 2018*) in control and *Graf*^CR57 embryos expressing Sqh-mCherry. As expected, *rok*^i expressing embryos showed diffuse distribution of Sqh-mCherry, loss of ring constriction and significantly larger ring area as compared to control embryos throughout cellularization (*Figure 9A, B, Video 10*). *rok*^i expression in the background of *Graf*^CR57 led to decrease in Sqh-mCherry recruitment and suppression of the constriction phenotype seen in *Graf*^CR57 embryos (*Figure 9A, B, Video 10*). Rho-GTP was likely to be unaffected in *rok*^i mutant embryos. Due to the absence of active Myosin II in the *Graf*^CR57; *rok*^i, these embryos showed a significantly higher ring area as compared to controls (*Figure 9A, B*). The Sqh-mCherry intensity in the ring compared to the inter ring in both *rok*^i and *Graf*^CR57; *rok*^i was decreased as compared to controls (*Figure 9C*).

Rho kinase also phosphorylates and deactivates Myosin II phosphatase (*Amano et al., 2010*; *Kimura et al., 1996*; *Mizuno et al., 1999*). Myosin II deactivation is defective in the mutants of Myosin II binding subunit (MBS) of Myosin II phosphatase (*Mizuno et al., 2002*; *Tan et al., 2003*). We expressed RNAi against MBS (*mbs*^i) (*Dey and Rikhy, 2020*; *Zhang et al., 2018*) maternally and stained embryos with fluorescently labeled phalloidin. As expected, *mbs*^i expression showed hyper constriction of the ring throughout cellularization. This hyper constriction phenotype in *mbs*^i embryos was suppressed by GRAF overexpression in maternally driven GRAF-OE embryos (*Figure 9—figure supplement 1*).

The suppression of *Graf* mutant phenotype of hyper constriction with Rok depletion shows that Myosin II activation is a necessary step to execute the ring constriction in cellularization (*Figure 9D*). In summary, our findings show that GRAF regulates the rate of contractile ring constriction by fine-tuning Rho-GTP levels at the furrow tip during mid cellularization.

## Discussion

In this study, we show that the function of the RhoGTPase domain containing protein GRAF is imperative in causing a delay in the onset of contraction during cellularization to ensure appropriate closure of cells at the base of nuclei after membrane extension. The spatial pattern of GRAF recruitment varies with the dynamics of the actomyosin network. GRAF localizes precisely to the furrow tip in early cellularization, increases in mid cellularization and finally becomes cytoplasmic at late stages (*Figure 9D*). *Graf* null mutant embryos show premature constriction in early cellularization and hyper constriction in mid and late stages. Depletion of *Graf* increases contractility and Myosin II levels, thereby leading to detachment of adjacent contractile units (*Figure 9D*). *Graf* loss leads to the tendency of the actomyosin network to attain a ring-like architecture prematurely and constriction is no longer restricted. The RhoGTPase domain deletion phenocopies the hyper constriction phenotype seen in *Graf* null mutant embryos. The spatial and temporal pattern of GRAF recruitment at the cleavage furrow along with its enzymatic activity make it a key protein in maintaining adhesion

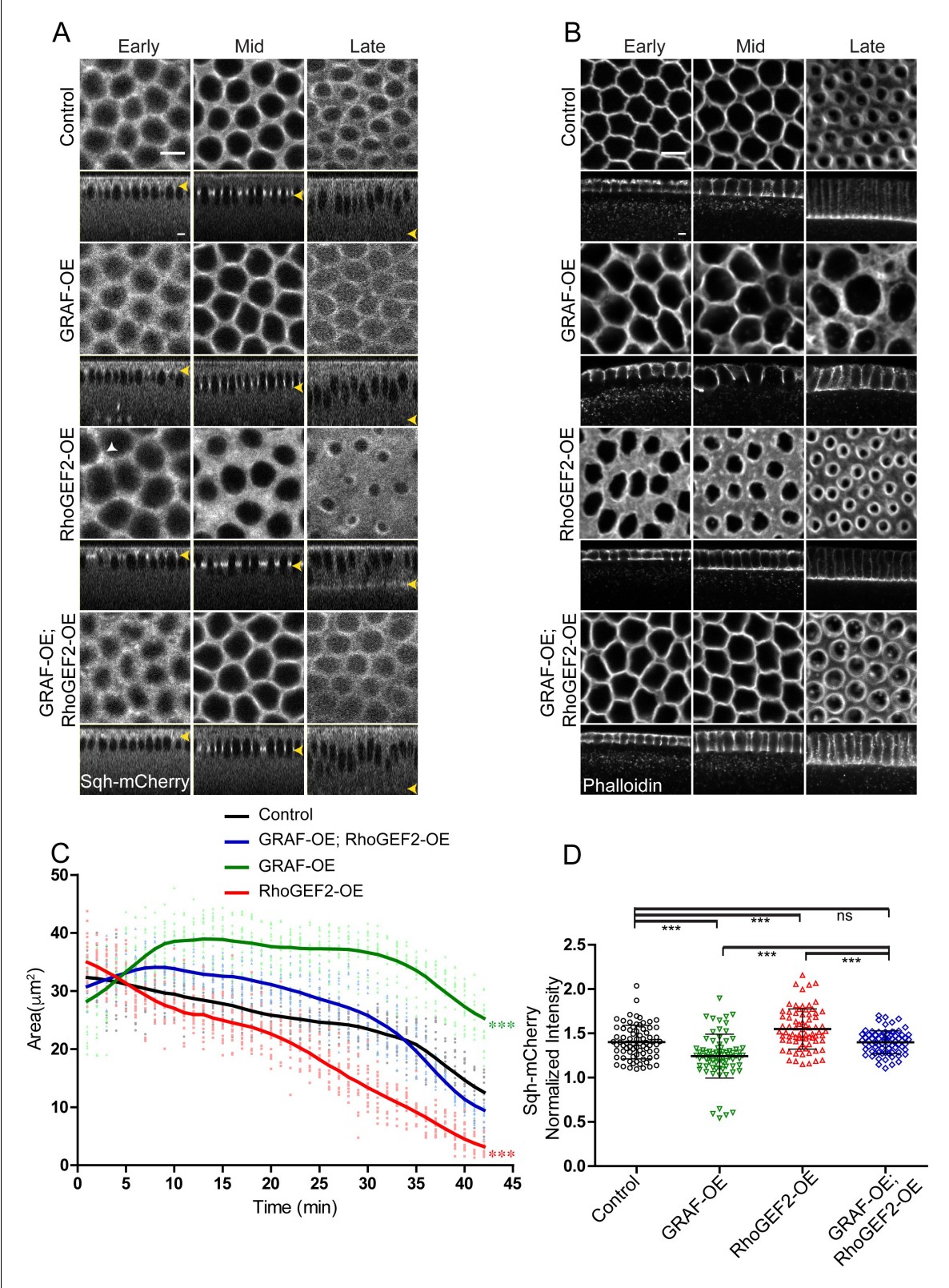

**Figure 7.** GRAF overexpression-driven loss of ring constriction is suppressed by RhoGEF2 overexpression in cellularization. (A–B) (A) The increased constriction seen in RhoGEF2-OE embryos is suppressed in GRAF-OE similar to controls when imaged live with Sqh-mCherry (gray) (yellow arrowhead marks furrow tip in sagittal sections, white arrowhead marks Sqh-mCherry foci in RhoGEF2-OE) and (B) when stained with phalloidin (B, GRAF-OE 100% shows loss of constriction, n = 50 embryos, RhoGEF2-OE 96.15% shows enhanced constriction, n = 52 embryos, GRAF-OE; RhoGEF2-OE 57.69% shows

*Figure 7 continued on next page*

*Figure 7 continued*

constriction comparable to controls, n = 52 embryos). (C) Quantification of contractile ring area from Sqh-mCherry shows decreased area in RhoGEF2-OE, increased area in GRAF-OE and normal area in GRAF-OE; RhoGEF2-OE as compared to controls (n = 15 rings, 5 rings per embryo, 3 embryos) (control values were repeated from *Figure 3* and GRAF-OE values were repeated from *Figure 4* for comparison). Data is represented as mean ± s.d. ***p<0.001, one-way ANOVA, repeated measure with Dunnett's multiple comparison test, the GRAF-OE, RhoGEF2-OE is statistically different from controls and GRAF-OE; RhoGEF2-OE. (D) Quantification of Sqh-mCherry in late cellularization in RhoGEF2-OE shows increased intensity as compared to controls, and GRAF-OE; RhoGEF2-OE shows Sqh-mCherry intensity a reduced intensity as compared to RhoGEF2-OE and is similar to controls (control and GRAF-OE values were repeated from *Figure 6* for comparison) (n = 75 rings, 5 rings per time point, 5 time points, 3 embryos each). Data is represented as mean ± s.d. ns: non-significant. ***p<0.001, two-tailed Mann–Whitney test. Scale bars: 5 µm.

The online version of this article includes the following source data and figure supplement(s) for figure 7:

**Source data 1.** Plotted values for ring area and fluorescence intensity analysis.

**Figure supplement 1.** RhoGEF2 overexpression retains GRAF recruitment and membrane furrow length analysis of RhoGEF2 depletion and overexpression.

**Figure supplement 1—source data 1.** Plotted values for furrow length analysis.

---

and inhibiting the contraction process in cellularization. Here, we discuss (1) the function of GRAF in spatiotemporal regulation of rate of constriction during cellularization, (2) coregulation of actomyosin assembly by actin remodeling or endocytic proteins along with GRAF and (3) redundancy of Rho-GAPs in morphogenesis in cellularization and gastrulation.

## Spatiotemporal recruitment of GRAF regulates the rate of constriction in cellularization

GRAF is present uniformly at the furrow tip when Myosin II is beginning to assemble in early stages of cellularization and a complete overlap between GRAF and Myosin II occurs during mid cellularization. Rho-GTP and Myosin II are present at the furrow tip in late cellularization, whereas GRAF is lost from the furrow tip in late stages. Constriction of the cleavage furrow in cellularization occurs in successive steps by the action of Myosin II and F-actin polymerization (*Xue and Sokac, 2016*). Myosin II assembles and forms a polygonal network in early cellularization. This polygonal network is resilient to constriction on activation of Myosin II, and RhoGTPase activities are predicted to be important at this step in restricting ring constriction (*Krueger et al., 2019*). Myosin II recruits during its assembly in phase I and initiates constriction at the ring in phase II in cellularization (*Krueger et al., 2019*; *Xue and Sokac, 2016*). The F-actin-severing protein, Cofilin, F-actin crosslinker fimbrin and F-actin-stabilizing proteins Anillin and septin, Peanut, affect ring constriction to a greater extent as compared to Myosin II in late cellularization (*Krueger et al., 2019*; *Mavrakis et al., 2014*; *Xue and Sokac, 2016*). The spatial recruitment of GRAF in early and mid cellularization at the furrow allows for its RhoGTPase activity to control levels of Rho-GTP in these stages. Sustained Rho-GTP levels due to absence of Rho-GTP hydrolysis in *Graf*

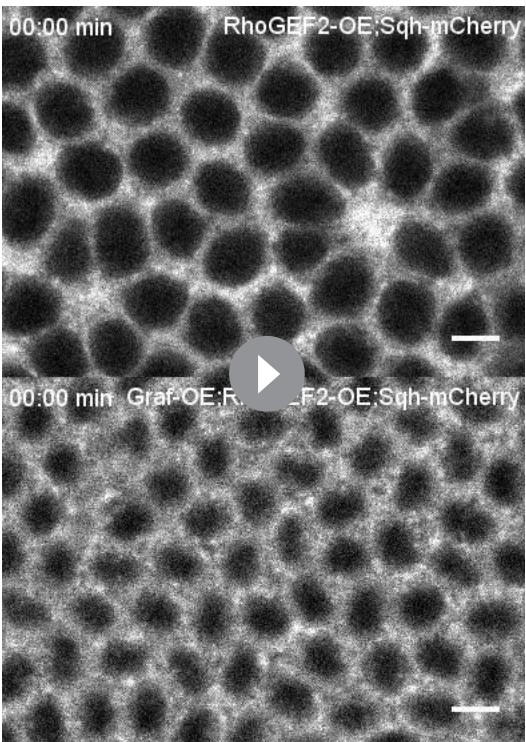

**Video 8.** The hyper constricted ring phenotype in RhoGEF2-OE expressing embryos is suppressed in GRAF-OE; RhoGEF2-OE. Constriction at the furrow was imaged during cellularization in RhoGEF2-OE (above) and GRAF-OE; RhoGEF2-OE (below) containing Sqh-mCherry (gray). RhoGEF2-OE shows hyper constriction in late cellularization, which is suppressed in the GRAF-OE;RhoGEF2-OE combination. Sum projections of five stacks at the furrow tip are shown at each time frame of the video. Scale bar = 5 µm.

https://elifesciences.org/articles/63535#video8

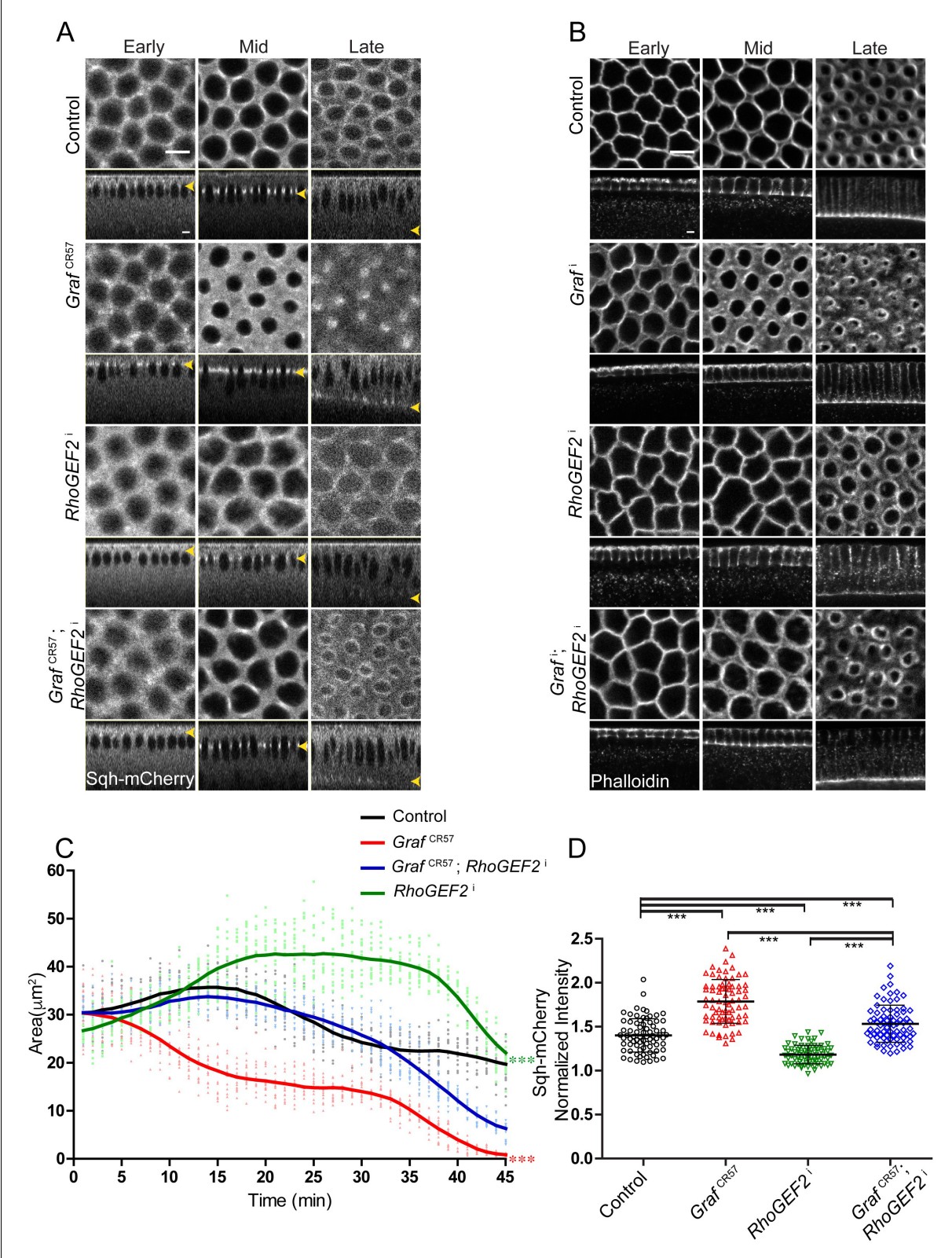

**Figure 8.** *Graf*^CR57 hyper constriction phenotype is suppressed by RhoGEF2 depletion. (A–B) *Graf*^CR57 (A) shows hyper constriction, *RhoGEF2*^i shows loss of constriction and *Graf*^CR57; *RhoGEF2*^i combination shows ring sizes comparable to control embryos when imaged live with Sqh-mCherry (gray) (yellow arrowhead denotes furrow tip). (B) *Graf*^i shows hyper constriction, *RhoGEF2*^i shows loss of constriction and *Graf*^i; *RhoGEF2*^i combination shows ring sizes comparable to control embryos when stained with fluorescent phalloidin (B, *Graf*^i 95.71% shows enhanced ring constriction, n = 70,

*Figure 8 continued on next page*

*Figure 8 continued*

*RhoGEF2*[i] 78.5% shows loss of constriction, n = 14, *Graf*[i]; *RhoGEF2*[i] 57.14%, rings comparable to controls, n = 21). (**C**) Quantification of contractile ring area from Sqh-mCherry expressing embryos shows decreased area in *Graf*[CR57], increased area in *RhoGEF2*[i] and rescued area in *Graf*[CR57]; *RhoGEF2*[i] (n = 15 rings, 5 rings per embryo, 3 embryos) (control values were repeated from *Figure 3* and *Graf*[CR57] values were repeated from *Figure 3* for comparison). Data is represented as mean ± s.d. ***p<0.001, one-way ANOVA, repeated measure with Dunnett's multiple comparison test, the *Graf*[CR57] and *RhoGEF2*[i] statistically different from controls and *Graf*[CR57]; *RhoGEF2*[i]. (**D**) Sqh-mCherry intensity in late cellularization in *RhoGEF2*[i] is decreased as compared to controls. *Graf*[CR57]; *RhoGEF2*[i] shows a decrease in Sqh-mCherry intensity as compared to *Graf*[CR57] and rescue in comparison with *RhoGEF2*[i] (control and *Graf*[CR57] values were repeated from *Figure 6* for comparison) (n = 75 rings, 5 rings per time point, 5 time points, 3 embryos each). Data is represented as mean ± s.d. ***p<0.001, two-tailed Mann–Whitney test. Scale bars: 5 μm.

The online version of this article includes the following source data and figure supplement(s) for figure 8:

**Source data 1.** Plotted values for ring area and fluorescence intensity analysis.
**Figure supplement 1.** RhoGEF2 depletion retains GRAF recruitment.

mutant embryos lead to increased Myosin II recruitment and, along with the F-actin remodeling activities of Anillin and Peanut, results in enhanced ring constriction in cellularization. Taken together, our study shows that spatiotemporal recruitment of GRAF to the contractile ring is required for fine-tuning of Rho activity for regulated constriction during cellularization.

## Role of actin regulatory and endocytic proteins in restriction of contractility in cellularization

Whereas Anillin and Peanut lead to constriction of the contractile ring in cellularization, Cheerio and Bottleneck are actin crosslinkers that inhibit contractility similar to GRAF. Bottleneck and Cheerio organize a basal polygonal non-contractile actin network in mid cellularization (*Krueger et al., 2019*; *Reversi et al., 2014*; *Schejter and Wieschaus, 1993b*; *Thomas and Wieschaus, 2004*). GRAF localization to this network plays a crucial role in limiting actomyosin contractility via its RhoGTPase activity. This is consistent with the previously characterized localization of GRAF on actin stress fibers and focal adhesions in mammalian cells (*Taylor et al., 1999*; *Taylor et al., 1998*). It is possible that Bottleneck and Cheerio mutants regulate actin polymerization downstream of Rho activation in cellularization. Bottleneck has also been shown to bind preferentially to phosphatidylinositol 3,4,5 trisphosphate (PIP3). It is interesting to note that an increase in phosphatidylinositol 4,5 bisphosphate (PIP2) causes hyper constriction during cellularization (*Reversi et al., 2014*). The PH and BAR domains of GRAF interact strongly with PIP2-containing liposomes (*Lundmark et al., 2008*). It is therefore possible that GRAF depletion leads to stabilization of PIP2 and loss of Bottleneck at furrow tips. Future studies on the analysis of Rho-GTP levels in Bottleneck and Cheerio mutants along with recruitment of Bottleneck and Cheerio in *Graf* mutants will reveal their interaction during actomyosin network stabilization in cellularization.

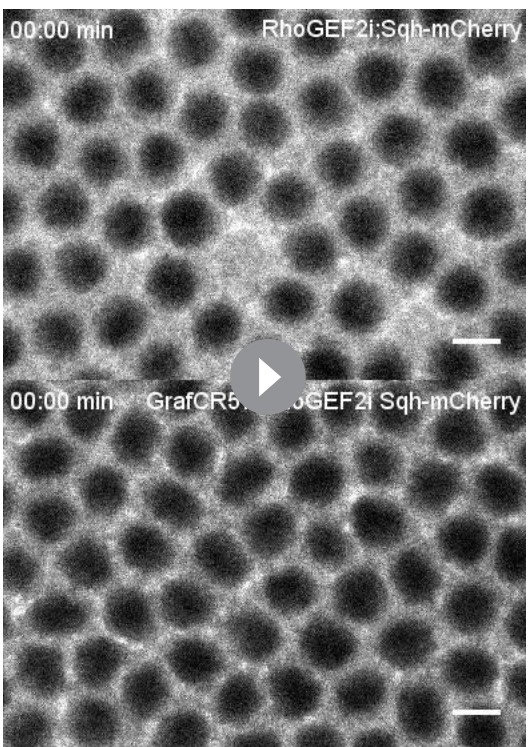

**Video 9.** Inhibition of ring constriction in *RhoGEF2*[i] expressing embryos is suppressed in *Graf*[CR57]; *RhoGEF2*[i]. Constriction at the furrow was imaged during cellularization in *RhoGEF2*[i] (above) and *Graf*[CR57]; *RhoGEF2*[i] (below) embryos expressing Sqh-mCherry (gray). *RhoGEF2*[i] knockdown embryos showed inhibition of ring constriction, and this phenotype is suppressed by *Graf*[CR57]; *RhoGEF2*[i]. Sum projections of five stacks at the furrow tip are shown at each time frame of the video. Scale bar = 5 μm.
https://elifesciences.org/articles/63535#video9

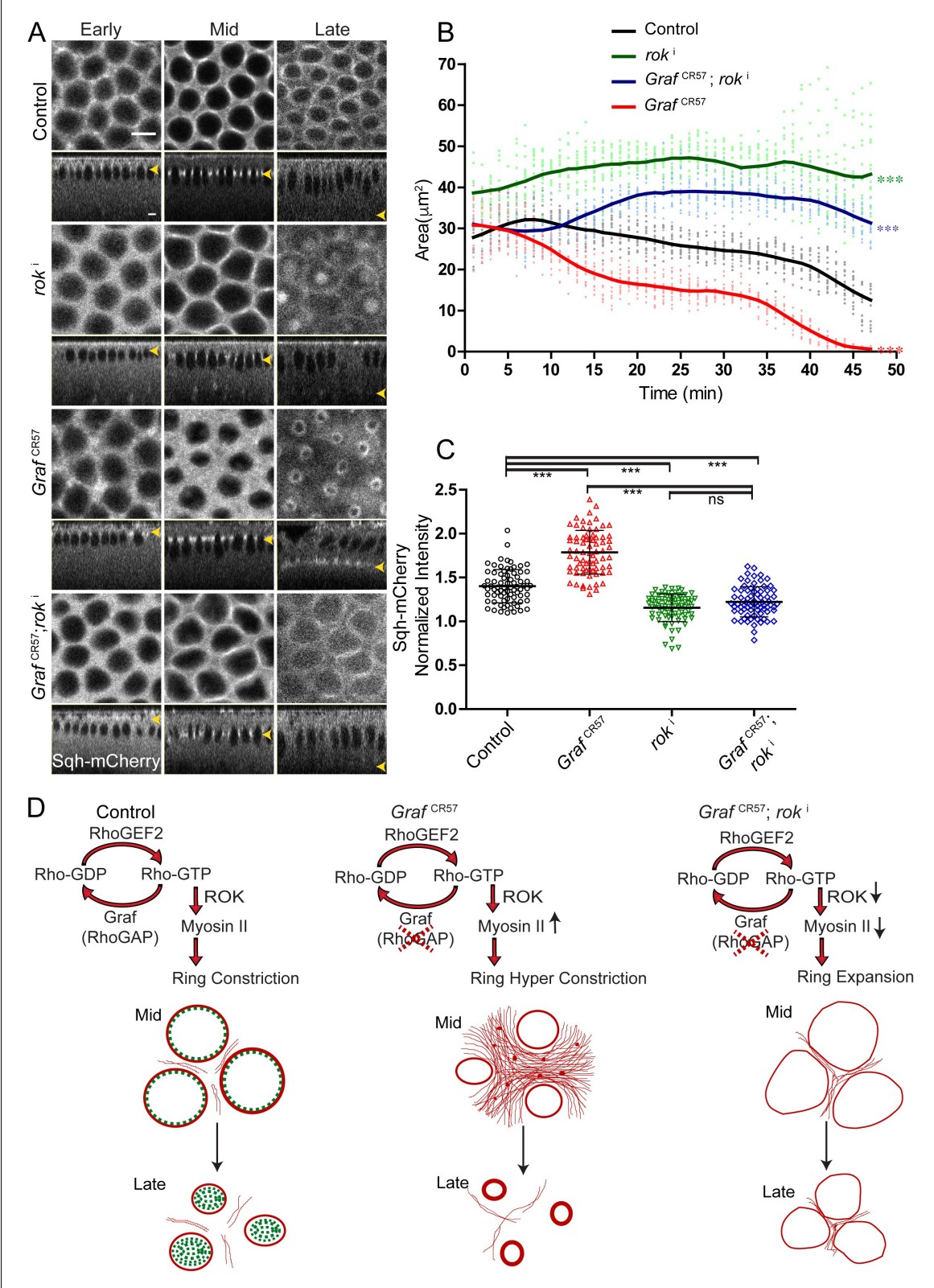

**Figure 9.** The hyper constriction phenotype of *Graf*CR57 embryos is suppressed by additional depletion of Rok. (**A**) *Graf*CR57 shows hyper constricted contractile rings, *rok*i shows loss of ring constriction and the *Graf*CR57; *rok*i combination suppression of the hyper constriction phenotype seen in *Graf*CR57 when imaged live with Sqh-mCherry (gray) (yellow arrowhead denotes furrow tip). (**B**) Quantification of contractile ring area from Sqh-mCherry embryos shows decreased area in *Graf*CR57, increased area in *rok*i and increased area in *Graf*CR57;*rok*i compared to controls (n = 15 rings, 5 per embryo,

*Figure 9 continued on next page*

*Figure 9 continued*

3 embryos) (control values were repeated from *Figure 3* and *Graf*^CR57 values were repeated from *Figure 3* for comparison). Data is represented as mean ± s.d. ***p<0.001, one-way ANOVA, repeated measure with Dunnett's multiple comparison test, the *Graf*^CR57, *rok*^i and *Graf*^CR57;*rok*^i ring area are statistically different from controls. (C) Quantification of Sqh-mCherry intensity in late cellularization in *rok*^i and *Graf*^CR57;*rok*^i shows decreased intensity in comparison to controls (control and *Graf*^CR57 values were repeated from *Figure 6* for comparison) (n = 75 rings, 5 rings per time point, 5 time points, 3 embryos each) (C). Data is represented as mean ± s.d. ns: non-significant. ***p<0.001, two-tailed Mann–Whitney test. Scale bars: 5 µm. (D) Schematic shows that GRAF plays a role in regulating Rho-GTP levels as a RhoGAP. GRAF is uniformly present at the contractile ring during mid stage and becomes cytosolic in late stages to drive the contraction process. Graf depletion shows a hyper constriction phenotype, and Rok depletion suppresses the hyper constriction phenotype seen in *Graf* mutant embryos. *Graf* mutant embryos show Myosin II accumulation at the ring in mid and late cellularization and in the inter ring region in mid cellularization.

The online version of this article includes the following source data and figure supplement(s) for figure 9:

**Source data 1.** Plotted values for ring area and fluorescence intensity analysis.

**Figure supplement 1.** GRAF overexpression suppresses the hyper constriction phenotype in Myosin II binding subunit (MBS)-depleted embryos.

GRAF functions in clathrin-independent endocytosis of the EGFR receptor in hematopoiesis in *Drosophila* (*Kim et al., 2017*). The small GTPase Arf1 recruits the human homologue of GRAF, Arh-GAP10, in clathrin-independent endocytosis of GPI-anchored proteins (*Kumari and Mayor, 2008*). Interestingly depletion of *Drosophila* Arf-GEF, *steppke,* leads to decreased endocytosis at the furrow and enhanced Rho1-induced hyper constriction in cellularization (*Lee and Harris, 2013*). Even though GRAF has been found to play a role in endocytosis in other systems, GRAF did not show vesicular staining during cellularization in our study. However, GRAF and ArfGEF may work either in similar or parallel pathways to regulate endocytosis at the cleavage furrow and in turn inhibit Rho1 activity during constriction.

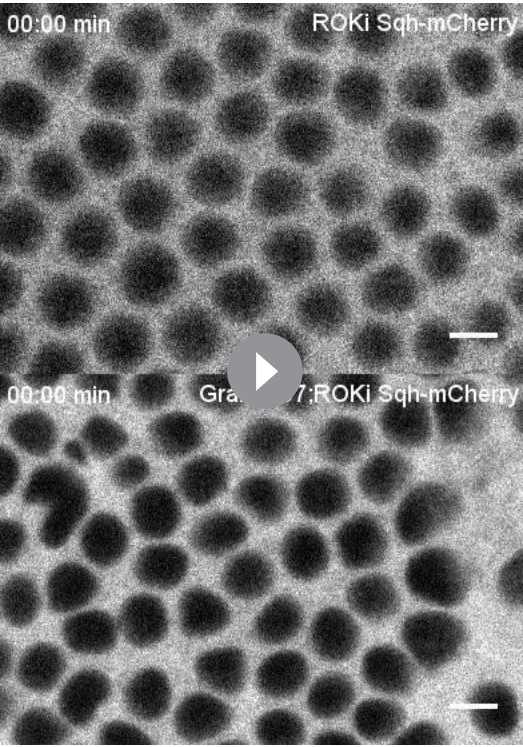

**Video 10.** *ROK*^i and *Graf*^CR57; *ROK*^i expressing embryos show inhibition of ring constriction. *ROK*^i (above) and *Graf*^CR57; *ROK*^i (below) embryos expressing Sqh-mCherry (gray) were imaged live during cellularization. *ROK*^i embryos show inhibition of ring constriction as compared to controls (*Video 1*). *ROK*^i embryos show accumulation of Sqh-mCherry at the cortex and below the nuclei. The decreased ring constriction seen in *ROK*^i and the hyper constriction phenotype seen in *Graf*^CR57 (*Video 5*) embryos is suppressed in the *Graf*^CR57; *ROK*^i combination. Sum projections of five stacks at the furrow tip are shown at each time frame of the video. Scale bar = 5 µm.
https://elifesciences.org/articles/63535#video10

## RhoGAP redundancy during cellularization and gastrulation

GRAF null mutant embryos give escaper adults during development. This suggests that there are other RhoGAPs that regulate Rho-GTP levels in the absence of GRAF in cellularization. Rho-GAP92B is another uncharacterized BAR, Rho-GAP and SH3 domain containing protein expressed in early embryogenesis (*Contrino et al., 2012*). CGAP is required for Rho-GTP cycling to produce Myosin II pulses for apical constriction during gastrulation in *Drosophila*. CGAP mutant embryos also show premature and hyper constriction during cellularization (*Mason et al., 2016*). In addition, RhoGAP68F is also required for coordinated apical constriction during gastrulation (*Sanny et al., 2006*). It is likely that GRAF, RhoGAP92B, CGAP and RhoGAP68F together regulate ring constriction during cellularization and apical constriction during gastrulation. CGAP depletion in *Graf* mutant embryos may enhance the hyper constriction phenotype seen in cellularization. This redundancy of function in the RhoGAPs appears

as a general feature and is seen in many organisms. RGA3 and RGA4 are examples of two RhoGAPs that regulate RhoA activity in early embryogenesis in *C. elegans* (*Schmutz et al., 2007*).

Multidomain proteins like GRAF containing membrane binding and RhoGAP domains are poised to regulate Rho-GTP levels in a highly regulated manner spatially and temporally during actomyosin contractility in different systems during cell migration and morphogenesis. Their presence in multi-cellular organisms and localization at focal adhesions and polarized epithelia suggests that they will regulate morphogenetic transitions occurring due to a change in adhesion and contractility.

# Materials and methods

### Key resources table

| Reagent type (species) or resource | Designation | Source or reference | Identifiers | Additional information |
|---|---|---|---|---|
| Genetic reagent (*Drosophila melanogaster*) | Canton-S | Bloomington *Drosophila* Stock Center | BDSC:1 RRID:BDSC_1 | |
| Genetic reagent (*Drosophila melanogaster*) | *nanos*-Gal4 | Lab stock (*Mavrakis et al., 2009*; *Mavrakis et al., 2008*) | | |
| Genetic reagent (*Drosophila melanogaster*) | w; *mat67*-Gal4; *mat15*-Gal4 | Girish Ratnaparkhi, IISER, Pune, India | | |
| Genetic reagent (*Drosophila melanogaster*) | w; P{Sqh-mCherry.M}3 | Bloomington *Drosophila* Stock Center | BDSC:59024 RRID:BDSC_59024 | |
| Genetic reagent (*Drosophila melanogaster*) | y[1] v[1];P{TRiP.HMC03427}attP40 (GRAFshRNA1) | Bloomington *Drosophila* Stock Center | BDSC:51853 RRID:BDSC_51853 | |
| Genetic reagent (*Drosophila melanogaster*) | y[1] v[1]; P{TRiP.GL01207}attP40 (myosin binding subunit, MBS shRNA) | Bloomington *Drosophila* Stock Center | BDSC:41625 RRID:BDSC_41625 | |
| Genetic reagent (*Drosophila melanogaster*) | y[1] sc[*] v[1] sev[21]; P{TRiP.HMS01118}a ttP2 (RhoGEF2 shRNA) | Bloomington *Drosophila* Stock Center | BDSC:34643 RRID:BDSC_34643 | |
| Genetic reagent (*Drosophila melanogaster*) | y[1] w[*];P{UASpT7. RhoGEF2}5 (RhoGEF2 overexpression) | Bloomington *Drosophila* Stock Center | BDSC:9386 RRID:BDSC_9386 | |
| Genetic reagent (*Drosophila melanogaster*) | *ubi*-GFP::AnillinRBD/TM3 | Thomas Lecuit, IBDM, France, *Munjal et al., 2015* | | |
| Genetic reagent (*Drosophila melanogaster*) | UASp-rok-shRNA (*rok*[i]) | Tony Harris, University of Toronto, Canada, *Zhang et al., 2018* | | |
| Genetic reagent (*Drosophila melanogaster*) | y[1] w[67]c[23] P{EPgy2}Graf[EY094 61] (GRAF overexpression) | Bloomington *Drosophila* Stock Center | BDSC:17571 RRID:BDSC_17571 | |
| Genetic reagent (*Drosophila melanogaster*) | w;*mat67* Spider-GFP-Sqh-mcherry/TM3ser | Eric F. Wieschaus, Princeton University, USA, *Martin et al., 2009* | | |
| Genetic reagent (*Drosophila melanogaster*) | y[1] sc[*] v[1] Graf[CR57]/FM7a (*Graf*[CR57]) | This study (Richa RIkhy, IISER, Pune, India) | | |
| Genetic reagent (*Drosophila melanogaster*) | y[1] sc[*] v[1] sev[21]; P{TKO.GS00762}att P40 | Bloomington Drosophila Stock Center | BDSC:76993 RRID:BDSC_76993 | |
| Genetic reagent (*Drosophila melanogaster*) | P{KK102763}VIE-260B (GRAF shRNA2, *Graf*[2i]) | Vienna *Drosophila* Stock Center, Vienna, Austria | v110812 | |

*Continued on next page*

*Continued*

| Reagent type (species) or resource | Designation | Source or reference | Identifiers | Additional information |
|---|---|---|---|---|
| Genetic reagent (*Drosophila melanogaster*) | y[1] sc[*] v[1] sev[21]; P{y[+t7.7] v[+t1.8]= nosCas9.R}attP40 | Bloomington *Drosophila* Stock Center | BDSC:78781 RRID:BDSC_78781 | |
| Genetic reagent (*Drosophila melanogaster*) | [w]*;p[UASp-GRAF-EGFPG1] attp40/cyo | This paper (Richa Rikhy, IISER, Pune, India) | | |
| Genetic reagent (*Drosophila melanogaster*) | [w]*;p[UASp-GRAFΔ RhoGAP-GFP] attp40/cyo | This paper (Richa Rikhy, IISER, Pune, India) | | |
| Sequence-based reagent | GRAF_pUASp_Homo_ Kpn1_GRAF(start)_FP (1) | This paper (Sigma-Aldrich) | PCR primers | CCGCATAGGCCACTAGTGGATCTGGTA CCATGGGCGGCGGCAAAAATGTACG |
| Sequence-based reagent | GRAF(end)_serine linker_GFP(start)_FP (2) | This paper (Sigma-Aldrich) | PCR primers | ACTATGTGGAACATTTGAAGCCGCACCA TTCCTCGAGCTCCAGCATGGTGAGCAAG GGCGAGGAGCT |
| Sequence-based reagent | GRAF(end)_serine linker_GFP(start)_RP (3) | This paper (Sigma-Aldrich) | PCR primers | AGCTCCTCGCCCTTGCTCACCATGCTGGA GCTCGAGGAATGGTGCGGCTTCAAATG TTCCACATAGT |
| Sequence-based reagent | GFP_pUASp_Homo_RP (4) | This paper (Sigma-Aldrich) | PCR primers | AACGTTCGAGGTCGACTCTAGAGGATCCTTA CTTGTACAGCTCGTCCATGCCGAGAGTGAT |
| Sequence-based reagent | RhoGAP_del_RP (5) | This paper (Sigma-Aldrich) | PCR primers | CCTTCGTGGCGTCCGGCAACTTTGCCTCG CTGACTTTGATTTTGCCG |
| Sequence-based reagent | RhoGAP_del_FP (6) | This paper (Sigma-Aldrich) | PCR primers | CGGCAAAATCAAAGTCAGCGAGGCAAAGT TGCCGGACGCCACGAAGG |
| Sequence-based reagent | GRAF FL AB vecOH FP (7) | This paper (Sigma-Aldrich) | PCR primers | ACGAAAATCTGTATTTCCAAGGCATGGGCGG CGGCAAAAATGT |
| Sequence-based reagent | Vect GRAF FL ABOH RP (8) | This paper (Sigma-Aldrich) | PCR primers | ACATTTTTGCCGCCGCCCATGCCTTGGAAAT ACAGATTTTCGT |
| Sequence-based reagent | Vect GRAF FL ABOH FP (9) | This paper (Sigma-Aldrich) | PCR primers | AACATTTGAAGCCGCACCATTGGTCT CATCCTCAGTTCGA |
| Sequence-based reagent | GRAF FL AB vecOH RP (10) | This paper (Sigma-Aldrich) | PCR primers | TCGAACTGAGGATGAGACCAATGGT GCGGCTTCAAATGTTCCAC |
| Sequence-based reagent | GRAF mutant scr1 FP (11) | This paper (Sigma-Aldrich) | PCR primers | GTAAATGTTGCAAACACCGCAGTTTT CTCGAAACTCAACC |
| Sequence-based reagent | GRAF mutant scr RP (12) | This paper (Sigma-Aldrich) | PCR primers | TACTACTTACTCTTTGCCGCACTCATA AGATCTTTGACCT |
| Sequence-based reagent | Pot2 212 CtoT FP (13) | This paper (Sigma-Aldrich) | PCR primers | CAATTGGCCATTTTACTTAACGATTTT |
| Sequence-based reagent | Pot2 212 CtoT RP (14) | This paper (Sigma-Aldrich) | PCR primers | AAAATCGTTAAGTAAAATGGCCAATTG |
| Sequence-based reagent | Pot2 420 AtoG FP (15) | This paper (Sigma-Aldrich) | PCR primers | AAGAAAACAAAAAGAAATTTGATAAAA |
| Sequence-based reagent | Pot2 420 AtoG RP (16) | This paper (Sigma-Aldrich) | PCR primers | TTTTATCAAATTTCTTTTTGTTTTCTT |
| Sequence-based reagent | Pot2 468 delT FP (17) | This paper (Sigma-Aldrich) | PCR primers | GCAGGAGCGTTTCCTCAATATGTC |
| Sequence-based reagent | Pot2 468 delT RP (18) | This paper (Sigma-Aldrich) | PCR primers | GACATATTGAGGAAACGCTCCTGC |
| Antibody | Anti-GRAF (rabbit polyclonal) | Bioklone | This study | 1:500; pre-absorbed using *Graf*[CR57] null mutant embryos |
| Antibody | Anti-Dlg (mouse monoclonal) | DSHB | 4F3, RRID:AB_528203 | IF (1:100) |
| Antibody | Anti-Zipper (rabbit polyclonal) | Thomas Jeffrey, Texas Tech University, TX, USA, **Chougule et al., 2016** | | IF (1:200) |

*Continued on next page*

*Continued*

| Reagent type (species) or resource | Designation | Source or reference | Identifiers | Additional information |
|---|---|---|---|---|
| Antibody | Goat anti-mouse IgG Alexa 488 | Molecular Probes | A-1100, RRID:AB_2534069 | IF (1:1000) |
| Antibody | Goat anti-rabbit IgG Alexa 488 | Molecular Probes | A-11008, RRID:AB_143165 | IF (1:1000) |
| Antibody | Goat anti-mouse IgG Alexa 568 | Molecular Probes | A-11004, RRID:AB_141371 | IF (1:1000) |
| Antibody | Goat anti-rabbit IgG Alexa 568 | Molecular Probes | A-1101, RRID:AB_143157 | IF (1:1000) |
| Antibody | Hoechst 33258 | Molecular Probes | H-3569 | IF (1:1000) |
| Antibody | Slow Fade Gold | Molecular Probes | S-36937 | |
| Antibody | Phalloidin Alexa 488 | Molecular Probes | A-12379, RRID:AB_2315147 | IF (1:100) |
| Antibody | Phalloidin Alexa 568 | Molecular Probes | A-12380, RRID:AB_2759224 | IF (1:100) |
| Antibody | Phalloidin Alexa 647 | Molecular Probes | A-22287, RRID:AB_2620155 | IF (1:100) |
| Software, algorithm | ImageJ and Fiji | National Institutes of Health, (*Rueden et al., 2017*; *Schindelin et al., 2012*; *Schneider et al., 2012*) | | |
| Software, algorithm | GraphPad Prism 5 | Prism | | |

## *Drosophila* stocks and genetics

Fly stocks and crosses were maintained in regular cornmeal agar. The detailed genotypes, stock numbers, source of stocks and temperature of crosses are a part of the Key resources table and *Supplementary file 1*.

## Generation of GRAF null mutant using the Crispr-Cas9 method

Flies containing *nos*-Cas9 were crossed to ubiquitous expressing gRNA against the first exon of GRAF to obtain F1 males containing *nos*-Cas9/GRAF gRNA (*Ren et al., 2013*; *Zirin et al., 2020*). In the germline of F1 males, the Cas9 enzyme cleaved the first exon of GRAF, leading to non-homologous end joining (NHEJ) repair and the occurrence of different mutations (*Figure 1—figure supplement 2B, C*). These males were crossed with FM7a virgin females. 155 F2 female progeny containing the mutant chromosome and the FM7a balancer were crossed individually to FM7a males. The crosses were checked for male lethality, and non-lethal stocks were tested for reduced progeny. One line was male lethal, several lines contained reduced progeny and several lines did not carry any remarkable phenotype. Genomic DNA was extracted from homozygous flies in lines having optimum progeny and those containing reduced progeny compared to control. Exon 1 of Graf was amplified from genomic DNA using the primers (Key resources table). The PCR product was detected by agarose gel electrophoresis and purified before sending it for sequencing to assess the presence of mutations in exon 1. Homozygous flies from 21 lines were sequenced, of which 18 lines contained reduced progeny and 3 lines did not show any phenotype (*Figure 1—figure supplement 2C*). 100% of the lines sequenced gave mutations in exon 1 of the *Graf* gene. Homozygous flies from lines with reduced progeny showed mutations in the *Graf* gene that led to a premature stop codon. Homozygous flies from lines that did not give any phenotype also gave mutations in exon, which did not alter the reading frame of the gene and were presumably not crucial for GRAF function. *Graf*^CR57 homozygous flies were maintained as a stock, and embryos from these flies were used for analysis of ring area, Myosin II and GRAF levels.

## Generation of GRAF-GFP and GRAFΔRhoGAP-GFP transgene

The *Graf* cDNA sequence was extracted from the LD28528 pOT2 vector (BDGP, USA) using primers 1 and 3 (Key resources table). The cDNA in LD28528 contained three mutations causing premature

stop codon when compared to the gene sequence available at FLYBASE:FBgn0030685. These mutations were rectified by site-directed mutagenesis as follows: C212T, A420G and deletion of T468 residue using primers 13–18 (Key resources table). The modified residues were confirmed using sequencing. This intact *Graf* cDNA sequence was used to generate a fluorescently tagged GRAF-GFP construct with GFP at the C-terminus. PCR amplification was performed on the GRAF and GFP gene using primers 1, 3 and 2, 4, respectively (Key resources table). Overlap PCR was performed to give rise to GRAF with a serine linker followed by GFP tagged at C-terminal of the *Graf* cDNA-amplified products using primers 1 and 4 (Key resources table). The PCR product at the end has a sequence homologous to BamH1 digested pUASp vector ends. The purified product from the overlap PCR was digested with BamH1. The pUASp vector was used in a 10:1 ratio for transformation in DH10B-derived *Escherichia coli* strain PPY (*Zhang et al., 2014*; *Zhang et al., 2012*). This strain recombines the pUASp vector and insert using the *in-bacto* homologous recombination strategy. PCR was performed to confirm the gene in the vector, and further the DNA sequence was confirmed by sequencing using primers 1 and 4 (Key resources table).

The deletion of the RhoGAP domain of GRAF that spans the residues 1177–1764 in the ORF was achieved using overlap PCR. PCR amplification was performed on pUASp GRAF-GFP vectors using primers 1,5 and 6,4. The overlap PCR was performed to give rise to GRAFΔRhoGAP with a serine linker followed by GFP tagged at C-terminal region of *Graf* cDNA-amplified products using primers 1 and 4 (Key resources table). This purified product had an end homologous region to BamH1, and both the PCR product and the pUASp vector were individually digested with BamH1. The purified product and vector were used in a 10:1 ratio to transform in *E. coli* strain PPY. Finally, primers 1 and 4 (Key resources table) were used for confirming the product with PCR, and the DNA sequence was confirmed by sequencing.

## Generation of anti-GRAF antibody

The pUASP GRAF-GFP vector was used to amplify GRAF full-length sequence using primers 7 and 10 (Key resources table), and pET-15b expression vector amplification was carried out by primers 8 and 9 (Key resources table). The GRAF full-length amplified product and vector were transformed into *E. coli* strain PPY in 10:1 ratio. PCR and sequencing were performed to confirm the presence of the GRAF insert using primers 7 and 10 (Key resources table). The N-terminal region of the GRAF gene had 6X His tag, which was used for protein purification.

Polyclonal antibodies were developed against full-length GRAF protein in New Zealand White rabbit at Bioklone, India, by the following procedure. Immunizations were given subcutaneously. After primary immunization of the rabbit with 200 g of GRAF in Freund's adjuvant (Sigma), boosters were administered with 150 g of GRAF at 21-day intervals. Production bleeds were collected from the immunized rabbit on the 14th day after each booster. The antibody titers in rabbit sera were measured in indirect ELISA. 100 µl of purified GRAF protein was coated in the wells of microtiter plates or strips (Nunc maxisorp) at a concentration of 3 g/ml, overnight at 4°C. After blocking with 5% skimmed milk in PBS for 1 hr at 37°C, the wells were washed twice with PBS. Varying dilutions of rabbit sera were added to the wells and incubated overnight at 4°C. After washes with PBS-Tween 20 (PBST) followed by PBS, the wells were incubated with goat anti-rabbit IgG-HRP (Merck, India), 1:2000, for 45 min at room temperature. Following washes with PBST and PBS, wells were incubated with TMB (Invitrosense ultra blue) at room temperature for 20 min in dark. After the addition of the stop solution (2N $H_2SO_4$), optical density (OD) values were measured at 450 nm using 800 TS absorbance reader (BioTek, Winooski, VT). Antibodies from the sixth bleed were purified on Protein A agarose beads (Sigma).

GRAF antibody staining on *Drosophila* embryos was standardized using various fixation protocols involving paraformaldehyde and heptane fixation along with methanol devitellinization and heat fixation. It was found that GRAF antibody staining was the best on heat fixation, and all experiments for GRAF staining were therefore performed with heat fixation.

## Immunostaining

0–3.5 hr embryos were collected on sucrose agar plates, washed, dechorionated with 100% bleach and washed again. To visualize GRAF, Zipper and Dlg immunostained embryos, dechorionated embryos were heat fixed with boiled 1X Triton salt solution (10x-0.5%Triton X-100% and 7% NaCl in

water) for 1 min and instantly adding ice-cold 1X washing buffer. After cooling on ice, embryos were devitelinized in 1:1 mix of MeOH and heptane followed by three washes in 1X PBST (1X PBS with 0.3% Triton X-100) for 5 min each. After washing, embryos were blocked in 2% BSA in 1X PBST for 1 hr and then incubated in the primary antibody (Key resources table) overnight at 4°C. This was followed by three 1X PBST washes and incubated in fluorescently coupled secondary antibodies (Molecular Probes) at 1:1000 dilution for 1 hr at room temperature. Embryos were washed three times in 1X PBST for 5 min each. DNA was labeled with Hoechst 33258 (1:1000, Molecular Probes) for 5 min in the second 1X PBST wash. Finally, embryos were mounted in Slow fade Gold antifade (Molecular Probes).

To visualize F-actin, embryos were fixed using 1:1 mix of 8% PFA in 1X PBS:heptane for 20 min and followed by hand devitellinization. Fluorescently coupled phalloidin (Molecular Probes) was used to label F-actin in embryos in cellularization for 1 hr at room temperature followed by washes and DNA labeling.

## Live imaging of *Drosophila* embryos

For live imaging, 2–2.5 hr embryos were collected on sucrose agar plates and dechorionated with 100% bleach for 1 min, and mounted on two-well coverslip bottom Labtek chambers. Mounted embryos were filled with 1X PBS (*Mavrakis et al., 2008*) and imaged using 40X/1.4NA oil objective on Zeiss or Leica SP8 microscope with a frame rate of 1.74 s/frame and 2 s/frame, respectively.

## Microscopy

Zeiss laser scanning confocal microscope LSM710, LSM780 and Leica laser scanning microscope SP8 containing laser lines at 488, 561 and 633 nm were used to image immunostained fixed or live embryos. The 40X objective having NA 1.4 of these microscopes was used for imaging. The laser power, scan speed and gain were adjusted with the range indicator mode such that 8-bit image acquisition was in 0–255 range. For both fixed or live imaging, an averaging of 2 was used during image acquisition. Optical sectioning of 1.08 µm and 0.68 µm was used to acquire images at Zeiss and Leica confocal microscopes, respectively.

## Laser ablation

Control and *Graf*[CR57] embryos expressing Sqh-mCherry were used for visualizing contractile rings for laser ablation experiments using the Zeiss LSM780 microscope. Laser ablation was achieved by using 800 nm multiphoton femtosecond pulsed Mai Tai laser. The region of interest used for ablations was set to a line of 510 pixel length (42.51 µm) and captured with a speed of 1.58 µs per pixel and 20 iterations. The Sqh-mCherry was imaged using a 561 nm laser excitation with a time interval of 1.27 s. A sagittal section was taken before ablations to estimate furrow length for early (less than 6 µm) and mid stages (6–16 µm) of cellularization. Three sections were taken before the ablations in approximately 3 s, the time taken for ablations is approximately 3.22 s and imaging was carried out for approximately 70 s after ablations.

## Image quantification and analysis

### Quantification of mean fluorescence intensity of GRAF and Zipper from immunostaining

The imaging of fixed control and mutant embryos immunostained with antibodies against GRAF along with Dlg (*Figure 2B, C*, *Figure 2—figure supplement 1B, C*) and Zipper along with Dlg (*Figure 6B*) was carried out on the Leica SP8 confocal microscope at the same laser power and gain settings. To estimate the mean fluorescence intensity per pixel of GRAF and Zipper from immunostaining, a single optical section from the Z-stack containing the brightest intensity at the furrow tip was chosen. Fluorescence intensity was obtained in this section by drawing ROIs (regions of interest) using the segmented line tool on the ring to get the mean intensity per pixel using Fiji software (http://fiji.sc/wiki/index.php/Fiji) (*Rueden et al., 2017*). The ring intensity obtained was divided by the mean cytosol intensity per pixel obtained from a large square ROI in the apical-most region above the nuclei from the same image. This cortex to cytosol ratio for GRAF antibody fluorescence in control embryos (*Figure 2B*), *Graf*[CR57] (*Figure 2C*), GRAF-OE (*Figure 5C*) and Zipper antibody fluorescence in control and Graf[CR57] (*Figure 6B*) was plotted for different stages of cellularization.

## Quantification of mean fluorescence intensity of GRAF, Sqh-mCherry and AnillinRBD-GFP from live imaging

Movies from live imaging of embryos containing GRAF-GFP,GRAFΔRhoGAP-GFP, Sqh-mCherry in GRAF-OE and $Graf^{CR57}$ (*Figure 4—figure supplement 1*, *Figure 6—figure supplement 1B*) were used to quantify the fluorescence intensity change with respect to time. Images with Z projection of sum intensity per pixel were obtained from two stacks above and two stacks below of the brightest section at the furrow tip (total five stacks) covering depth of 4 µm. ROIs across the furrow tip from 5 or 10 rings were drawn in these images for each time point to obtain the mean signal intensity per pixel. The mean intensity obtained at each time point was represented as a ratio to the maximum mean fluorescence value per pixel obtained across cellularization within each embryo and finally plotted as a 'normalized intensity versus time.'

A line ROI passing through the ring for a single optical plane was used to obtain an intensity profile of Sqh-mCherry, GRAF-GFP and AnillinRBD-GFP in embryos across different stages of cellularization (*Figure 2E*, *Figure 4B,D*). The intensity at the documented time points was represented as a ratio to the maximum intensity usually across cellularization for each of these embryos.

Live imaging of Sqh-mCherry expressing embryos in various genotypes was used to quantify fluorescence intensity change during the last five time points in late cellularization. Sqh-mCherry showed an increased accumulation at rings in late cellularization in $Graf^{CR57}$ as compared to controls. To estimate these changes quantitatively across all genotypes, images were obtained with sum intensity per pixel across the Z axis from a total of five stacks: two stacks above and two stacks below of the brightest section at the furrow tip covering depth of 4 µm in late cellularization. The mean ring intensity per pixel was extracted by drawing a segmented ROI on the ring. The inter ring intensity per pixel was extracted by drawing an ROI in between adjacent rings that had reduced Sqh-mCherry intensity. The ring intensity per pixel was expressed as a ratio to inter ring intensity per pixel. Finally, the normalized intensity of the ring was shown as a scatter plot (*Figure 6D*, *Figure 7D*, *Figure 8D*, *Figure 9C*).

## Quantification of contractile ring area

Phalloidin-stained embryos marking contractile rings at the basal-most region were extracted for quantification of contractile ring area from control, $Graf^{i}$ and $Graf^{CR57}$ (*Figure 1E*). These images were converted to 8-bit and transformed to binary images. The binary images were inverted and segmented to identify individual rings, and the area inside the rings was quantified for different stages of cellularization. The area was plotted in groups of early (furrow length less than 6 µm), mid (furrow length 6–16 µm) and late (furrow length more than 16 µm) stages of cellularization based on furrow length to compare between control and mutant embryos.

Live movies from embryos expressing Sqh-mCherry of different genotypes were used to quantify contractile ring area in *Figure 4G*, *Figure 5E*, *Figure 7C*, *Figure 8C* and *Figure 9B*. Sqh-mCherry fluorescence images containing sum intensity were obtained across five optical sections at the base of the furrow with the brightest section in the middle. Five rings per embryo were marked manually using a polygon tool in these images, and the area inside the ring was computed in ImageJ. The mean ± s.d. for the area was computed and plotted with time using GraphPad Prism 5.0.

## Quantification of furrow membrane length

Sqh-mCherry fluorescence was used to identify furrow tips during cellularization. Membrane length was measured using the ImageJ line tool after every 2 min in *Figure 6—figure supplement 1A* and *Figure 7—figure supplement 1B* (five furrow lengths per time point were recorded in each embryo). These lengths were plotted against time as a scatter plot using GraphPad Prism 5.0.

## Quantification of displacement and initial recoil velocity after laser ablation

The manual tracking plugin of Fiji software was used to extract xy coordinates from control and ablated regions. These xy coordinates were used to get distance between the two points with the following formula:

$$D = \sqrt{(y2 - y1)^2 + (x2 - x1)^2}$$

where (x1,y1) and (x2,y2) are the coordinates of the contractile ring edge seen below and above ablation region, respectively.

The ring displacement was estimated for each time point after ablations. The displacement of edges measured before ablations was subtracted from all the time points after ablations. The displacement was calculated from five independent embryos. The mean displacement and s.d. with respect to time was plotted for control and *Graf*^CR57 mutant embryos across the early and mid stages (*Figure 3B, F*). The displacement at the last time point at approximately 70 s was used to plot maximum displacement (*Figure 3C, G*). The initial recoil velocity was calculated by fitting a linear function on points between 0 and 23 s on the plot of displacement versus time in each embryo (*Figure 3D, H*).

## N values and statistical analysis

For area quantification, the mean ± s.d. was computed from a total of 60 rings from six embryos of each genotype in fixed and 15 rings from three embryos each genotype in live embryos. For intensity quantification, the n-value for each experiment is mentioned in their respective figure legends. The statistical significance was determined using non-parametric Mann–Whitney test, two-tailed Student's t-test to compare two means (*Figure 1E*, *Figure 2B*, *Figure 3C, D, G, H*, *Figure 5C,E*, *Figure 6B,D*, *Figure 7D*, *Figure 8D*, *Figure 9C*, *Figure 2—figure supplement 1B*, *Figure 4—figure supplement 1B*). One-way ANOVA, repeated measures was used to compare the area curves to check for statistical significance between various genotypes followed by Dunnett's multiple comparison test (*Figure 4G*, *Figure 7C*, *Figure 8C*, *Figure 9B*). Two-way ANOVA was also used to compare the statistical difference at each time point between genotypes in *Figure 6—figure supplement 1B*. Graphs for area versus time (*Figure 4G*, *Figure 5E*, *Figure 7C*, *Figure 8C*, *Figure 9B*) and intensity versus time (*Figure 4—figure supplement 1B*, *Figure 6—figure supplement 1B*) were represented with smoothing using the second-order, seven-neighbors 'Savistsky-Golay' smoothing algorithm.

## Acknowledgements

We thank RR lab members for continuous discussions on the data in this work. We thank the *Drosophila* and Microscopy facility at IISER, Pune, India. Stocks from the Bloomington *Drosophila* Stock Center were used for this study. We thank Vimlesh Kumar from IISER, Bhopal, India, for providing *Graf* cDNA for cloning. We thank K Rajeshwari from Bioklone for GRAF antibody generation. We thank NCBS *Drosophila* injection facility for embryo injections to generate transgenics. We thank ImageJ, Fiji and Flybase for image analysis and genetic analysis used in the paper. We thank Thomas Jeffrey (Texas Tech University, TX, USA) for the anti-Zipper antibody. RR thanks the Department of Biotechnology, India, and IISER, Pune, India, for funding for this project. SS thanks Council of Scientific and Industrial Research, India, for graduate fellowship.

## Additional information

### Funding

| Funder | Grant reference number | Author |
| --- | --- | --- |
| Department of Biotechnology | BT/PR26071/GET/119/108/2017 | Richa Rikhy |
| Council of Scientific and Industrial Research, India | Graduate fellowship | Swati Sharma |

The funders had no role in study design, data collection and interpretation, or the decision to submit the work for publication.

### Author contributions

Swati Sharma, Conceptualization, Resources, Formal analysis, Validation, Investigation, Visualization, Methodology, Writing - original draft, Writing - review and editing; Richa Rikhy, Conceptualization, Resources, Formal analysis, Supervision, Funding acquisition, Validation, Investigation, Visualization, Methodology, Writing - original draft, Project administration, Writing - review and editing

## Author ORCIDs

Richa Rikhy (ID) https://orcid.org/0000-0002-4262-0238

## Decision letter and Author response

Decision letter https://doi.org/10.7554/eLife.63535.sa1
Author response https://doi.org/10.7554/eLife.63535.sa2

## Additional files

### Supplementary files

• Supplementary file 1. *Drosophila* recombinants and crosses. The stocks and recombinants used in this study are numbered in Table 1. The crosses carried out with the stocks for each figure for live imaging and fixed imaging of embryos along with the temperature at which each cross is carried out are detailed in Table 2.

• Transparent reporting form

### Data availability

The data generated and analysed during this study are included in the source data files.

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
