## [Decision Letter]

**Acceptance summary:**

This paper will be of general interest to the cell biology community as it focuses on the basic mechanisms of Rho signaling and cellular contractility. The data presented here support the claims by the authors in showing that in the absence of the Rho-GAP scaffolding protein GRAF, developing *Drosophila* embryos experience hyper-contractility during cellularization. Careful analyses of this hyper-contractility phenotype were performed including laser ablation experiments which connect GRAF expression to the regulation of Rho signaling.

**Decision letter after peer review:**

Thank you for submitting your article "Spatiotemporal recruitment of RhoGTPase protein GRAF inhibits actomyosin ring constriction in *Drosophila* cellularization" for consideration by *eLife*. Your article has been reviewed by 3 peer reviewers, including Derek Applewhite as the Reviewing Editor and Reviewer #1, and the evaluation has been overseen by Anna Akhmanova as the Senior Editor. The following individual involved in review of your submission has agreed to reveal their identity: Jordan Beach (Reviewer #3).

The reviewers have discussed the reviews with one another and the Reviewing Editor has drafted this decision to help you prepare a revised submission.

Summary:

In this manuscript Sharma and Rikhy show that actomyosin contraction at invaginating furrows during *Drosophila* cellularization is modulated by the RhoGAP GRAF. The authors demonstrate that GRAF localizes at the tip of the invaginating furrows and that its depletion or over-expression causes changes in the dynamics of ring contraction. The authors then argue that GRAF affects myosin II localization and activity by balancing the levels of Rho-GTP at the contractile rings. This study carefully dissects the role of GRAF during cellularization and nicely shows its importance to achieve the correct levels of actomyosin contraction during this process. Due to the central role of actomyosin mechanics in several morphogenetic processes, as well as in disease, this work is potentially relevant for a wide audience. Furthermore, the authors have developed an arsenal of tools to explore the role of GRAF that should be useful to the fly community. However, I believe that some issues should be addressed before publication particularly regarding the quantification and interpretation of the data.

Essential revisions:

1. The authors provide a number of data points consistent with hypercontractility but never demonstrate that the ring is actually hypercontractile. Ideally laser ablations to analyze recoil, or experiments using tension sensors could be performed. However, RLC phosphorylation could suffice. Analysis of phosphorylated light chains with/without GRAF help support the model that the myosin being recruited is active/contractile.

2. Validation of KD lines. For some, I could be convinced that previous literature is sufficient (eg Rok KD), but for GRAF, which is central in this work, I would like to see in author's hands that they are seeing protein depletion, what the maternal contribution level is, etc. Similarly, the authors do an excellent job of presenting their CRISPR KO validation up to the point of showing loss of protein. I have seen examples where N-terminal targeted CRISPR KO lines were bypassed by unknown alternative starts. It would be nice to demonstrate that no shorter proteins are being produced in the KO, as far as the author's antibodies can determine.

3. The authors lean heavily on the area measurements as a readout for rescues (GFP-GRAF +/- GAP) and genetic interactions (ROCK, RhoGEF2, etc). More quantitative investigation of these rescues would be appropriate. The authors have the tools to do this. For example, the zipper (or RLC-P as suggested above) recruitment would be a nice downstream readout to confirm the area differences are myosin 2 dependent.

4. GRAF-OE (with or without Rho-GEF2) appears to have more lateral actin. Were the images in Figure 6 collected and presented identically? Acquisition and post-acquisition modifications were identical? The question regarding acquisition/post-acquisition modifications to the images applies throughout the paper. It is not clear to me if the reader can directly compare intensities between early/mid/late samples or if they were all adjusted independently for presentation. Please make this clear. In a number of the images, the noise appears to increase in Late, is this because there is more diffuse GRAF or because the laser/signal was adjusted to enhance the image for presentation?

5. In figure 3C the authors claim that in the GRAFdeltaRhoGAP-GFP embryos the phenotype is similar to that in GRAF^CR57^. However, I am not convinced of this based on the data. Is the initial area of the rings significantly different between control embryos and GRAFdeltaRhoGAP-GFP? In that case, are the dynamics of ring contraction over time different, or simply rings are contracting at similar rates and are smaller in GRAFdeltaRhoGAP-GFP because their original size was smaller? Initially, the dynamics of contraction in GRAF^CR57^embryos seems significantly faster than in GRAFdeltaRhoGAP-GFP, which could imply that the RhoGAP domain of GRAF is important at later stages of cellularization. Differences in dynamics would be more obvious if the curves were normalized to the initial size of the rings. The authors should look at the rate of contraction to be able to claim changes in contractility speed.

6. For your graphs measuring the change in ring area of over time (Figures 3C, 4E, 5B, 6C, 7C, 8B) an ANOVA analysis has been performed on each time point or the overall rate? The figure legend indicates these are statistically significant, but there is no indication in the figure. It is just hard to follow exactly what is statistically significant in these graphs.

7. The authors are looking at the mean fluorescence of different proteins like myosin and GRAF. Although this is a very informative metric, I think they should also look at the total levels of these molecules to see if they are increasing or not. For example, the authors say that the levels of GRAF and myosin increase from early to mid-cellularization. However, usually mean fluorescence is normalized to (divided by) the area of the structure under study. As the rings are decreasing in area during cellularization, the increase in mean fluorescence they are observing could simply be an increase in the density of myosin or GRAF as the rings become smaller. Therefore to talk about an increase in the recruitment of these molecules, the authors should look at total fluorescence levels (the product of mean fluorescence and area of the ring). This is particularly important when comparing the levels of molecules in phenotypes where the ring sizes differ at a given time point, like figure 5D.

8. Figure 8: The lower diagram in C shows Graf and ROK KD leads to "Ring constriction similar to controls". This is not supported by the data. You've broken the chain below the GAP, and therefore the phenotype should mimic ROK-KD alone, which is what 8B shows. The blue line is more similar to the green than to the black. ROK isn't in the competitive cycle like RhoGEF2 is. It is downstream. Regardless of what happens upstream, ROK-KD should largely prevent myosin activation and constriction, which the authors data show. And "Dephosphorylated myosin 2" would not lead to "Ring Constriction similar to controls" as the model suggests, it would lead to the results presented in the data.

---

## [Author Response]

Essential revisions:1. The authors provide a number of data points consistent with hypercontractility but never demonstrate that the ring is actually hypercontractile. Ideally laser ablations to analyze recoil, or experiments using tension sensors could be performed. However, RLC phosphorylation could suffice. Analysis of phosphorylated light chains with/without GRAF help support the model that the myosin being recruited is active/contractile.

We have attempted the following different experiments to address change of active Rho-GTP and contractility in the controls and GRAF mutant embryos:

1. As suggested by the reviewers, we performed laser ablations at the ring for controls and mutants to test changes in contractility in early and mid cellularization. We found that the initial recoil velocity, rate of displacement and displacement at the final time point of the ring at the furrow base is increased at early and mid stages of cellularization in Graf^CR57^ as compared to controls. This new experiment has been added as a part of revised Figure 3 and Video 3.

2.Since GRAF has a RhoGAP domain, we further tested the levels of Rho-GTP in mutant embryos. For this, we add new data of Rho-GTP distribution as visualized by AnillinRBD-GFP in *Graf*^CR57^ and controls. AnillinRBD-GFP is more spread in mid cellularization between adjacent rings in *Graf*^CR57^ as compared to controls. This allows a documentation of change in Rho-GTP levels in embryos in cellularization in *Graf*^CR57^. This is included as a part of revised Figure 4.

3. We have measured the changes in Myosin II recruitment to the contractile ring by estimating the intensity of Zipper immunofluorescence in fixed embryos and Sqh-mCherry in live videos in controls and mutants. We find that Sqh-mCherry and Zipper levels are increased in *Graf*^CR57^ as compared to controls. This is included as a part of revised Figure 6.

4. As suggested by the reviewers, we also attempted an analysis of phosphorylated light chain with a phosphospecific antibody against Sqh obtained from Robert Ward. We add images here for reference of the reviewers (Author response image 1). We tried various fixation methods to confirm this antibody: heat fixation, PFA fixation followed by hand devitellinization and PFA fixation followed by methanol devitellinization. We did not get phospho-Sqh staining similar to controls published in (Zhang and Ward 2011). We saw some patchy staining on increasing the laser power in controls at the furrow but did not see a significant difference of this staining between the control and mutant embryos. It may be possible that the antibody stability has been affected during the transport contributing to the loss in specific staining.

**Author response image 1. sa2fig1:** Sqh1P expression pattern during mid cellularization in *Graf*^CR57^ mutant. Sqh1P staining revealed patchy staining with cytoplasmic signal during mid cellularization. There is no specific localization at the furrow tip in contractile rings and no observable change in the Sqh1P pattern in *Graf*^CR57^ mutants as compared to controls.

5. We have not been able to test the tension sensor (Cai et al. 2014) in cellularization in *Drosophila* embryos but plan to do this in future experiments.

2. Validation of KD lines. For some, I could be convinced that previous literature is sufficient (eg Rok KD), but for GRAF, which is central in this work, I would like to see in author's hands that they are seeing protein depletion, what the maternal contribution level is, etc. Similarly, the authors do an excellent job of presenting their CRISPR KO validation up to the point of showing loss of protein. I have seen examples where N-terminal targeted CRISPR KO lines were bypassed by unknown alternative starts. It would be nice to demonstrate that no shorter proteins are being produced in the KO, as far as the author's antibodies can determine.

1. We used a guide RNA to target exon1 of the Graf gene. This gave rise to the *Graf*^CR57^ mutant by a premature stop codon which gave rise to a truncated product of 27aa. The reason to target exon1 instead of targeting the deletion of the entire gene was because of the presence of another gene CG8260 in the opposite direction from the intronic region of Graf. Even though CG8260 does not show the presence of transcripts in the development as per the ModEncode data base, we did not want to affect its synthesis by deleting the entire genomic region. This information has been added to the revised text on creation of Crispr mutants of *Graf*.

2. The *Graf*^CR57^ mutant embryos show loss of GRAF antibody staining and the mutant phenotype of hypercontractility is rescued by the addition of full length GRAF-GFP in embryos. This rescue is not seen when a deletion construct missing the RhoGAP domain is combined with *Graf*^CR57^ mutant embryos. This rescue shows a specific requirement of the RhoGAP activity of GRAF in inhibiting contractility in cellularization.

3. The Crispr mutant of *Graf* is partially viable and escapers can be used to maintain these homozygous flies as a stock. It does not show any defects in development of other organ structures such as the wing, eye, thoracic bristles, etc. We performed RNAi expression against GRAF with maternal, wing and eye specific Gal4s. These experiments showed a specific defect in embryonic stages with maternal Gal4s and not with wing specific or eye specific Gal4s or even tubulin Gal4. These data together suggest that maternal deposition and/or synthesis of GRAF is present in early development and is essential for these developmental processes.

4. We also show that there is a depletion of GRAF antibody staining in Graf RNAi expressing embryos. We have added this data to revised Figure 2—figure supplement 1. GRAF levels are decreased to a greater extent in *Graf*^CR57^ embryos as compared to Graf RNAi expressing embryos. This also correlates with the stronger ring constriction phenotype seen in Graf ^CR57^ as compared to Graf RNAi. Both the RNAi and the Crispr mutant show the loss of function, hyper constriction phenotypes. It is very unlikely that other products produced in the GRAF Crispr mutant affect the phenotype.

5. We attempted western blot analysis to test whether smaller fragments are seen in the *Graf*^CR57^ embryos. We get multiple bands in the wild type extracts from adults and embryos with the antibody against GRAF and the control blots with anti-tubulin antibody show a specific single band. The western blot analysis from mutants shows loss of protein bands in mutants in a couple of blots but this is not reproducible. As per the data on ModEncode and Fly Proteome, the GRAF mRNA and protein are present in very small quantities. It is likely that for this reason and possibly due to less stability of the protein, we are unable to detect this on western blots with specificity and reproducibility. Our current and future studies involve attempts to standardise conditions to increase the stability of the protein for biochemistry experiments.

6. For embryos expressing RNAi against RhoGEF2 and ROK and with overexpression of RhoGEF2 we include new measurements of Sqh-mCherry at late cellularization in revised Figures 6, 7, 8 and 9.

Sqh-mCherry shows a patchy distribution in early stages of cellularization during Myosin II assembly the furrow tip. It shows localization at the ring in mid and late stages of cellularization. The intensity in mid stages is higher than late stages of cellularization in controls. Sqh-mCherry is more spread in the inter ring region in Graf mutants in mid cellularization. This has also been seen in actin remodelling protein Cheerio mutant embryos which show hyper constriction (Krueger et al. 2019). Sqh-mCherry increased in intensity at the contractile ring in *Graf*^CR57^ and decreased in intensity in the inter ring region in late cellularization. We therefore quantified the ring to inter ring region ratio for all the genotypes from live videos with Sqh-mCherry in cellularization. This analysis shows that Sqh-mCherry fluorescence is reduced at the ring in RhoGEF2 and ROK RNAi expression embryos and increased in *Graf*^CR57^, *Graf*^CR57^;GRAFΔRhoGAP-GFP and RhoGEF2-OE embryos.

We have also measured the Sqh-mCherry and Zipper intensity in syncytial embryos in prophase and metaphase in RhoGEF2 RNAi, RhoGEF2-OE and ROK RNAi in previous studies (Dey and Rikhy 2020). Sqh-mCherry is present in the cortical regions close to the membrane in control embryos in prophase and is cytoplasmic in metaphase. We find that Sqh-mCherry and Zipper are cytoplasmic in prophase in ROK RNAi and RhoGEF2 RNAi expressing embryos and are retained in the cortical regions in metaphase of RhoGEF2-OE embryos as compared to controls. The reagents for knockdown and overexpression of Rho and Myosin II have therefore been confirmed in several ways of shape analysis and Myosin II intensity estimation in cellularization in the revised manuscript and in syncytial stages of development previously. We have added the appropriate description from literature for each of these mutants in the Results section.

3. The authors lean heavily on the area measurements as a readout for rescues (GFP-GRAF +/- GAP) and genetic interactions (ROCK, RhoGEF2, etc). More quantitative investigation of these rescues would be appropriate. The authors have the tools to do this. For example, the zipper (or RLC-P as suggested above) recruitment would be a nice downstream readout to confirm the area differences are myosin 2 dependent.

As mentioned in response to point 1, we did not get a significant staining for p-Sqh at the ring in controls and Graf ^CR57^ mutants. Hence we could not use this method to document changes of p-Sqh levels in single or double mutant embryos.

We performed Zipper antibody stainings in *Graf*^CR57^ mutant embryos and found increased levels in early, mid and late cellularization. Sqh-mCherry is recruited to the ring in late stages and is not present in inter-ring regions in control embryos. We found that Sqh-mCherry levels remained high at the ring in *Graf*^CR57^ mutant embryos in late cellularization. We therefore estimated the levels of Sqh-mCherry in Graf ^CR57^ in late cellularization at the ring relative to the inter ring from videos. We found an increase in Sqh-mCherry relative to inter ring regions in *Graf*^CR57^, *Graf*^CR57^;GRAFΔRhoGAP-GFP when compared to controls and *Graf*^CR57^; GRAF-GFP (Revised Figure 6).

We also estimated the levels of Sqh-mCherry as a ratio to the inter ring intensity in all the mutants used in the manuscript. This data is included as a part of revised Figures 7, 8 and 9. We found that there is an increase in Sqh-mCherry at the ring in RhoGEF2-OE and this was no longer seen in combinations of GRAF-OE; RhoGEF2-OE (Figure 7). We found that there is a decrease in Sqh-mCherry in RhoGEF2 RNAi expressing embryos and this was not seen in the combinations of *Graf*^CR57^;RhoGEF2 RNAi (Figure 8). A decrease in Sqh-mCherry was seen in ROK RNAi mutant embryos and this decrease remained in *Graf*^CR57^, ROK RNAi embryos thus confirming that hyper contractility in *Graf*^CR57^ embryos is dependent upon Myosin II activation via Rho-kinase (Figure 9).

4. GRAF-OE (with or without Rho-GEF2) appears to have more lateral actin. Were the images in Figure 6 collected and presented identically? Acquisition and post-acquisition modifications were identical? The question regarding acquisition/post-acquisition modifications to the images applies throughout the paper. It is not clear to me if the reader can directly compare intensities between early/mid/late samples or if they were all adjusted independently for presentation. Please make this clear. In a number of the images, the noise appears to increase in Late, is this because there is more diffuse GRAF or because the laser/signal was adjusted to enhance the image for presentation?

The immunostainings for GRAF-OE are imaged at similar laser power settings to controls on the confocal microscope. There is an increase in GRAF all along the Z axis in GRAF-OE embryos (Figure 5). As noted by the reviewer, this also coincides with an increase in Phalloidin and therefore increase in F-actin in GRAF-OE embryos. It may be possible that an increase in GRAF gives rise to an increase in F-actin along the furrow.

We are currently investigating the change in recruitment of actin remodelling proteins in *Graf* mutant embryos and will be able to highlight the impact of GRAF on regulating actin remodelling in a future study.

5. In figure 3C the authors claim that in the GRAFdeltaRhoGAP-GFP embryos the phenotype is similar to that in GRAF ^CR57^. However, I am not convinced of this based on the data. Is the initial area of the rings significantly different between control embryos and GRAFdeltaRhoGAP-GFP? In that case, are the dynamics of ring contraction over time different, or simply rings are contracting at similar rates and are smaller in GRAFdeltaRhoGAP-GFP because their original size was smaller? Initially, the dynamics of contraction in GRAF ^CR57^ embryos seems significantly faster than in GRAFdeltaRhoGAP-GFP, which could imply that the RhoGAP domain of GRAF is important at later stages of cellularization. Differences in dynamics would be more obvious if the curves were normalized to the initial size of the rings. The authors should look at the rate of contraction to be able to claim changes in contractility speed.

The quantification of area for *Graf*^CR57^; GRAFΔRhoGAP-GFP and *Graf*^CR57^; GRAF-GFP has now been redone using the Sqh-mCherry localization as a readout for making the ROI for estimating area. The earlier distribution was estimated using the GFP fluorescence of GRAF-GFP or GRAFΔRhoGAP-GFP, this tends to label the edges better than the vertices and often appears to be rounded. This gave an underestimate of the area especially in early cellularization. The revised Figure 4 shows that there is a decrease in area in *Graf*^CR57^, GRAFΔRhoGAP-GFP similar to *Graf*^CR57^ when compared to both controls and in *Graf*^CR57^, GRAF-GFP embryos throughout cellularization.

6. For your graphs measuring the change in ring area of over time (Figures 3C, 4E, 5B, 6C, 7C, 8B) an ANOVA analysis has been performed on each time point or the overall rate? The figure legend indicates these are statistically significant, but there is no indication in the figure. It is just hard to follow exactly what is statistically significant in these graphs.

We have added a description of the curves that are statistically different in the figures and legends in the revised manuscript. One-way ANOVA with the modification used compares the overall curve for area change between different genotypes.

We have used the two-way ANOVA to compare each time point for Sqh-mCherry intensity between *Graf*^CR57^, GRAF-OE and controls in Figure 6—figure supplement 1B. The description for this has been added to the legends.

7. The authors are looking at the mean fluorescence of different proteins like myosin and GRAF. Although this is a very informative metric, I think they should also look at the total levels of these molecules to see if they are increasing or not. For example, the authors say that the levels of GRAF and myosin increase from early to mid-cellularization. However, usually mean fluorescence is normalized to (divided by) the area of the structure under study. As the rings are decreasing in area during cellularization, the increase in mean fluorescence they are observing could simply be an increase in the density of myosin or GRAF as the rings become smaller. Therefore to talk about an increase in the recruitment of these molecules, the authors should look at total fluorescence levels (the product of mean fluorescence and area of the ring). This is particularly important when comparing the levels of molecules in phenotypes where the ring sizes differ at a given time point, like figure 5D.

All the fluorescence intensities for the GRAF antibody, Zipper antibody, GRAF-GFP, AnillinRBD-GFP and Sqh-mCherry are represented as a mean per pixel at any given time point or cellularization stage in the previous and the revised manuscript.

We have shown line scans of the relative fluorescence intensity across a line with fixed thickness and length at the furrow tip for Sqh-mCherry, GRAF-GFP and AnillinRBD-GFP. We find that Sqh-mCherry and GRAF-GFP colocalize in early stages. GRAF-GFP increases at the furrow tip in mid stages and becomes cytoplasmic in late cellularization (revised Figure 2). Sqh-mCherry on the other hand increases in mid cellularization and is still present in the ring in late cellularization.

The mean fluorescence intensities per pixel for Sqh-mCherry are obtained from sum projections of 5 sections in revised Figure 6—figure supplement 1B. The mean fluorescence intensities per pixel are represented as a ratio to the maximum fluorescence per embryo. This shows that there is an increase in Sqh-mCherry and GRAF in every pixel in mid cellularization followed by a decrease in late cellularization. We have included this description in the revised Materials and methods.

The mean fluorescence intensities per pixel obtained from the sum intensity images relative to inter ring intensity per pixel in late stages are now added in Figures 6, 7, 8 and 9 to estimate the relative change in Myosin II recruitment in different mutant and rescue combinations. We have included this description in the revised material and methods.

8. Figure 8: The lower diagram in C shows Graf and ROK KD leads to "Ring constriction similar to controls". This is not supported by the data. You've broken the chain below the GAP, and therefore the phenotype should mimic ROK-KD alone, which is what 8B shows. The blue line is more similar to the green than to the black. ROK isn't in the competitive cycle like RhoGEF2 is. It is downstream. Regardless of what happens upstream, ROK-KD should largely prevent myosin activation and constriction, which the authors data show. And "Dephosphorylated myosin 2" would not lead to "Ring Constriction similar to controls" as the model suggests, it would lead to the results presented in the data.

We thank the reviewers for pointing this out. Yes this was indeed a typographical error. We have made a new summary diagram in Figure 9 and corrected this to show that there is loss of constriction in ROK RNAi and this is maintained in the *Graf*^CR57^ ROK RNAi combinations. ROK depletion suppresses the hyper constriction phenotype of *Graf*^CR57^.